# Targeted redox inhibition of protein phosphatase 1 by Nox4 regulates eIF2α-mediated stress signaling

Celio XC Santos[1], Anne D Hafstad[1,2], Matteo Beretta[1], Min Zhang[1], Chris Molenaar[1], Jola Kopec[1,3], Dina Fotinou[1,3], Thomas V Murray[1], Andrew M Cobb[1], Daniel Martin[1], Maira Zeh Silva[1,3], Narayana Anilkumar[1], Katrin Schröder[4], Catherine M Shanahan[1], Alison C Brewer[1], Ralf P Brandes[4], Eric Blanc[5], Maddy Parsons[3], Vsevelod Belousov[6], Richard Cammack[7], Robert C Hider[7], Roberto A Steiner[3] & Ajay M Shah[1,*]

## Abstract

**Phosphorylation of translation initiation factor 2α (eIF2α) attenuates global protein synthesis but enhances translation of activating transcription factor 4 (ATF4) and is a crucial evolutionarily conserved adaptive pathway during cellular stresses. The serine–threonine protein phosphatase 1 (PP1) deactivates this pathway whereas prolonging eIF2α phosphorylation enhances cell survival. Here, we show that the reactive oxygen species-generating NADPH oxidase-4 (Nox4) is induced downstream of ATF4, binds to a PP1-targeting subunit GADD34 at the endoplasmic reticulum, and inhibits PP1 activity to increase eIF2α phosphorylation and ATF4 levels. Other PP1 targets distant from the endoplasmic reticulum are unaffected, indicating a spatially confined inhibition of the phosphatase. PP1 inhibition involves metal center oxidation rather than the thiol oxidation that underlies redox inhibition of protein tyrosine phosphatases. We show that this Nox4-regulated pathway robustly enhances cell survival and has a physiologic role in heart ischemia–reperfusion and acute kidney injury. This work uncovers a novel redox signaling pathway, involving Nox4–GADD34 interaction and a targeted oxidative inactivation of the PP1 metal center, that sustains eIF2α phosphorylation to protect tissues under stress.**

**Keywords** eIF2α; metal center; Nox4; protein phosphatase; redox signaling
**Subject Categories** Autophagy & Cell Death; Signal Transduction; Structural Biology
**The EMBO Journal (2016) 35: 319–334**

## Introduction

Stresses such as protein misfolding in the endoplasmic reticulum (ER), nutrient deprivation, hypoxia, ischemia and oxidants trigger the reversible phosphorylation of serine 51 on the α-subunit of the eukaryotic initiation factor 2 (eIF2α) (Walter & Ron, 2011). eIF2α is part of the multimeric eIF2 complex that initiates mRNA translation (Baird & Wek, 2012). Phosphorylation of eIF2α inhibits translation and global protein synthesis but increases the cap-independent translation of certain mRNAs, such as activating transcription factor 4 (ATF4) (Vattem & Wek, 2004). ATF4 induces genes involved in amino acid transport and biosynthesis, stress resistance, and autophagy and contributes to cell survival during environmental stresses (Harding *et al*, 2003; Liu *et al*, 2008; Ye *et al*, 2010; Baird & Wek, 2012; Hart *et al*, 2012). Because diverse stress signals converge to trigger eIF2α phosphorylation and ATF4 translation, the pathway is known as the integrated stress response (ISR) (Harding *et al*, 2003).

Four mammalian kinases, namely protein kinase R, protein kinase R-like ER kinase (PERK), general control non-derepressible-2, and heme-regulated eiF2α kinase, couple different upstream stress signals to eIF2α phosphorylation (Baird & Wek, 2012). eIF2α is dephosphorylated by the serine–threonine protein phosphatase 1 (PP1), which is targeted to eIF2α by a PP1-binding protein, growth arrest and DNA damage-inducible 34 (GADD34) (Novoa *et al*, 2001). GADD34 itself is induced by ATF4, thereby acting as a brake on the ISR (Ma & Hendershot, 2003). Previous work shows that inhibition of eIF2α dephosphorylation by GADD34 deletion or small molecule inhibitors of the GADD34–PP1 complex prolongs eIF2α phosphorylation and increases cell survival during acute stresses (Kojima *et al*, 2003; Boyce *et al*, 2005; Tsaytler *et al*, 2011).

---

1 Cardiovascular Division, King's College London British Heart Foundation Centre of Excellence, London, UK
2 Cardiovascular Research Group, Department of Medical Biology, The Arctic University of Norway, Tromsø, Norway
3 Randall Division, King's College London British Heart Foundation Centre of Excellence, London, UK
4 Institute for Cardiovascular Physiology, Goethe-University, Frankfurt, Germany
5 MRC Centre for Developmental Neurobiology, King's College London, London, UK
6 Shemyakin-Ovchinnikov Institute of Bioorganic Chemistry, Moscow, Russia
7 Institute of Pharmaceutical Science, King's College London, London, UK
 *Corresponding author. Tel: +44 207848 5189; E-mail: ajay.shah@kcl.ac.uk

Cellular stress situations are typically associated with increased levels of reactive oxygen species (ROS). Mitochondrial ROS produced primarily through electron transport chain leak may change significantly depending on metabolic activity (Murphy, 2009). The ER is the site for oxidative protein folding, and increased client protein load is associated with higher ROS levels (Han *et al*, 2013). In contrast to such ROS production as a by-product of other cellular functions, ROS may also be generated as the primary function of NADPH oxidase (Nox) family enzymes (Bedard & Krause, 2007). Seven mammalian Noxs (Nox1-5 and Duox1-2) with tissue-specific expression are described, which generate ROS (superoxide $[O_2^-]$ or hydrogen peroxide $[H_2O_2]$) by catalyzing electron transfer from NADPH to molecular $O_2$. Noxs are particularly important in cell compartment-specific localized redox signaling in processes such as cell differentiation, proliferation, migration, and tissue repair (Bedard & Krause, 2007; Lassègue *et al*, 2012). Among the mammalian Noxs, Nox4 has the most extensive tissue distribution and has several unique properties of interest in relation to the ISR. Nox4 is found in the ER (in addition to other locations), is induced by stresses such as hypoxia, ischemia, ER stress, and mechanical stress, and has constitutive catalytic activity unlike other Noxs which require specific activation (Pedruzzi *et al*, 2004; Bedard & Krause, 2007; Wu *et al*, 2010; Zhang *et al*, 2010; Lassègue *et al*, 2012). Moreover, recent studies indicate that the upregulation of endogenous Nox4 mediates adaptive phenotypes in the heart, vasculature, and kidney under chronic disease stress (Zhang *et al*, 2010; Nlandu Khodo *et al*, 2012; Schröder *et al*, 2012), in contrast to Nox1 and Nox2 which mediate detrimental tissue remodeling (Lassègue *et al*, 2012). However, the precise role(s) of Nox4 induction during cellular stress and the molecular mechanisms through which it mediates signaling remain unclear.

Here, we show that Nox4 exerts cytoprotective effects by augmenting eIF2α phosphorylation and ATF4 levels during protein misfolding stress. Nox4 is upregulated downstream of ATF4 and interacts directly with GADD34 to form a macromolecular complex containing PP1 at the ER, thereby causing a local ROS-mediated inhibition of PP1 activity. Unlike the redox inhibition of protein tyrosine phosphatases (PTPs), which involves cysteine oxidation, PP1 inhibition by $H_2O_2$ involves oxidation of its metal center.

Nox4-mediated inhibition of PP1 resulting in enhanced eIF2α phosphorylation and increase in ATF4 levels increase cell survival during protein misfolding stress and is robustly protective against acute cardiac or kidney injury. Collectively, our results identify a previously unrecognized pathway that regulates eIF2α phosphorylation through a novel spatially localized oxidative inhibition of serine–threonine PP1 activity, and suggest new therapeutic targets to ameliorate acute cardiac and renal injury.

## Results

### Nox4 selectively regulates ATF4 during protein misfolding stress

Protein misfolding stress (ER stress) triggers the ISR through the activation of PERK (Walter & Ron, 2011). ER stress also activates IRE1 (inositol-requiring enzyme 1) and ATF6 (activating transcription factor 6), as part of the unfolded protein response (UPR). We first studied the effects of Nox4 on ER stress signaling in cardiac cells. Both H9c2 cells and primary cardiomyocytes exposed to an *N*-glycosylation inhibitor, tunicamycin (an established inducer of ER stress), showed a time- and dose-dependent increase in Nox4 protein and mRNA levels (Fig 1A; Appendix Fig S1A and B). Nox4 was also induced by thapsigargin, which evokes ER stress by depleting ER calcium (Fig 1B). The shRNA-mediated depletion of Nox4 significantly decreased basal and tunicamycin-stimulated increases in Nox activity in membrane fractions (Appendix Fig S1C and D), blunted tunicamycin-induced increases in the ER chaperones Grp78, Grp94, and calreticulin, and profoundly decreased ATF4 protein levels as compared to control cells (Fig 1C, Appendix Fig S1E). However, protein levels of cleaved ATF6 and mRNA levels of *Xbp1-s* (X-box binding protein 1, spliced form), as readouts of ATF6 and IRE1 signaling, respectively, were unaltered. Similar results were obtained after Nox4 knockdown with an independent siRNA (Appendix Fig S2A). Overexpression of Nox4 in H9c2 cells significantly increased basal and stress-induced levels of ATF4 and ER chaperones but had no significant effect on ATF6 levels, while *Xbp1-s* levels were slightly decreased (Fig 1D, Appendix Fig S2B). The alterations in ATF4 levels after manipulation of Nox4

---

**Figure 1.  Nox4 selectively regulates ATF4 during ER stress.**

A    Tunicamycin (Tn, 2 μg/ml) increased Nox4 protein levels in H9c2 cells. Tubulin was used as a loading control. *n* = 4–6/group. *, significant compared to baseline.

B    Thapsigargin (Tp, 1 μg/ml) increased Nox4 protein levels in H9c2 cells. *n* = 4–6/group. *, significant compared to baseline.

C    Effect of Nox4 on the unfolded protein response. Nox4 was depleted in H9c2 cells by shRNA-mediated knockdown (Ad.shNox4), or cells were treated with a control adenovirus (Ad.Ctl). In cells with Nox4 knockdown, tunicamycin treatment resulted in lower increases in protein levels of the ER chaperones Grp94, Grp78, and calreticulin than in control cells. Nuclear protein levels of ATF4 were substantially lower in Nox4-depleted cells than in control cells, but the levels of cleaved ATF6 (ATF6c) were similar. Histone was used as a loading control. The relative mRNA levels of *Xbp1-s* (a readout of IRE1 signaling) were unaltered after Nox4 knockdown. Mean data are shown in Appendix Fig S1E. Similar results were obtained with an independent siRNA approach (Appendix Fig S2A).

D    Effect of adenoviral-mediated overexpression of Nox4 (Ad.Nox4) or a control β-galactosidase protein (Ad.β-Gal) on tunicamycin responses of H9c2 cells. Nox4 enhanced the increase in cellular ER chaperones and nuclear ATF4 levels but did not affect tunicamycin-induced changes in nuclear ATF6c levels and caused minor reduction in *Xbp1s* mRNA levels. Mean data are shown in Appendix Fig S2B.

E, F    Effect of Nox4 knockdown or overexpression, respectively, on the tunicamycin-induced changes in mRNA levels of ATF4 target genes. *n* = 4/group. *Psat1*, phosphoserine aminotransferase; *Phgdh*, 3-phosphoglycerate dehydrogenase; *Asns*, asparagine synthetase; *Slc6a9*, glycine transporter 1.

G    Effect of ATF4 silencing with two different siRNAs on Nox4 protein levels in tunicamycin-treated H9c2 cells. Scrambled siRNAs were used as a control (Ctl). Representative immunoblots shown to the top (captions at bottom of bar graphs refer also to the immunoblots); tubulin was used as a loading control. *n* = 4/group. *, significant compared to baseline; #, significant comparing siATF4 versus corresponding siCtl.

H    Effect of ATF4 overexpression on *Nox4* mRNA and protein levels. *n* = 3/group.

Data information: All blots are representative of at least three independent experiments. Data are presented as mean ± SEM. Comparisons in (A, B) were made by one-way ANOVA and in other panels by Student's *t*-test. *P* < 0.05 was considered significant. Values above bar graphs denote the level of significance.

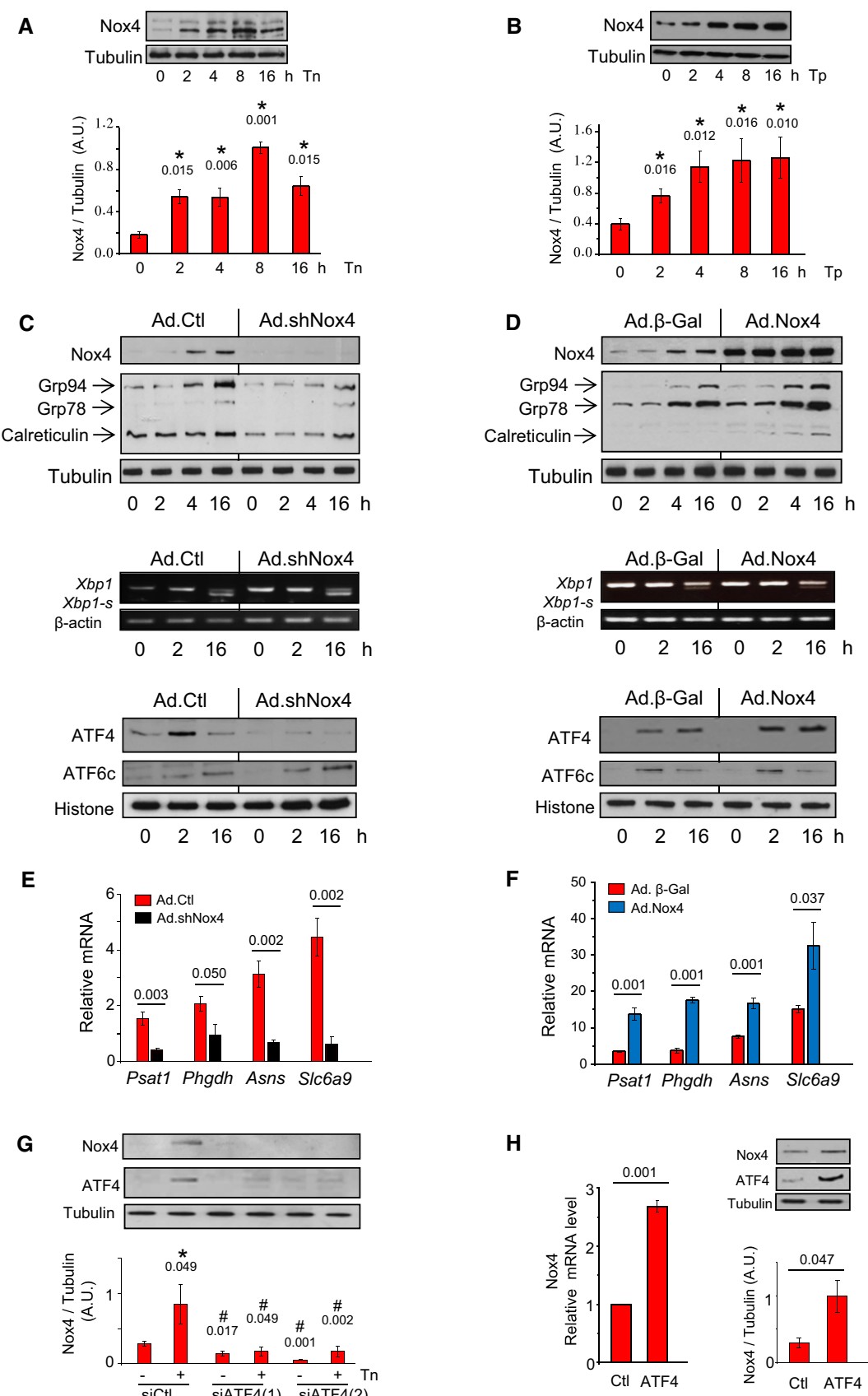

**Figure 1.**

expression were accompanied by corresponding changes in ATF4 target genes (Fig 1E and F).

These results indicate that Nox4 specifically regulates the ATF4 limb of the UPR during ER stress signaling but raise the question how Nox4 itself is upregulated. We found that the knockdown of ATF4 with two different siRNAs substantially blunted tunicamycin-induced increases in Nox4 protein and mRNA (Fig 1G, Appendix Fig S2C). Conversely, ATF4 overexpression induced significant increases in *Nox4* mRNA and protein (Fig 1H). Examination of the 10-kb rat genomic Nox4 promoter sequence proximal to the transcriptional start site, using MatInspector software (http://www.genomatix.de), identified three regions which comprise potential binding sites for ATF4, suggesting that *Nox4* might be a direct transcriptional target of ATF4. To test this possibility, we undertook chromatin immunoprecipitation (ChIP) assay. This showed that a region containing 2 canonical ATF4 binding motifs (−3,525 to −3,410, relative to the *Nox4* translational start site; Appendix Fig S2D) demonstrated binding to ATF4, which increased in the presence of tunicamycin. These data suggest that at least part of the increase in *Nox4* may involve direct *cis*-regulation by ATF4. Taken together, these results indicate a bidirectional positive signaling between Nox4 and ATF4.

### Nox4 inhibits eIF2α-related PP1 activity and enhances eIF2α phosphorylation

To establish how Nox4 regulates ATF4 levels, we first assessed eIF2α phosphorylation. In control H9c2 cells, tunicamycin-induced increase in eIF2α Ser51 phosphorylation peaked at 2 h and thereafter decreased (Fig 2A and C), a time-course attributable to progressive eIF2α dephosphorylation by the GADD34–PP1 complex. Nox4 knockdown markedly inhibited tunicamycin-induced eIF2α phosphorylation, whereas Nox4 overexpression led to sustained prolongation of stress-induced eIF2α phosphorylation (Fig 2A–D). Neither knockdown nor overexpression of Nox4 affected the levels of active phosphorylated PERK (Fig 2A and B, Appendix Fig S3A and F), suggesting that mechanisms downstream of PERK are involved in the modulation of eIF2α phosphorylation.

To assess the contribution of PP1-mediated eIF2α dephosphorylation, we first quantified GADD34 and PP1 protein levels. Nox4 knockdown was accompanied by a marked reduction in GADD34, but no change in PP1 levels (Fig 2A, Appendix Fig S3B and C). The lower GADD34 levels can be attributed to the reduction in ATF4 (Fig 1C) but should reduce PP1 activity and thereby increase eIF2α phosphorylation. Nox4 overexpression also had no effect on PP1 levels and slightly increased basal GADD34 levels (Fig 2B, Appendix Fig S3G and H), a pattern that again cannot account for the sustained eIF2α phosphorylation in this setting. We next assessed PP1 activity. In control cells, tunicamycin induced a significant time-dependent increase in phosphatase activity (Fig 2E and F). Nox4 knockdown resulted in significantly greater increases in phosphatase activity, whereas overexpression abrogated the increase in activity, suggesting that Nox4 augments phospho-eIF2α levels predominantly by inhibiting PP1 activity. We tested whether Nox4-dependent modulation of PP1 activity affects other PP1 target proteins in tunicamycin-treated H9c2 cells but found no changes in the pattern of either glycogen synthase or histone H3 phosphorylation (Fig 2G and H, Appendix Fig S3D, E, I and J).

We also tested the effects of Nox4 in mouse embryonic fibroblasts (MEFs), a cell type extensively employed in studies of conserved stress responses. Similar to cardiac cells, tunicamycin-treated Nox4$^{-/-}$ MEFs showed a significant blunting of the increases in phospho-eIF2α, ATF4, and ER chaperones observed in wild-type (WT) MEFs (Fig 2I, Appendix Fig S3K–M). The transfection of Nox4 into Nox4$^{-/-}$ MEFs reversed these changes but had no effect on the phosphorylation of glycogen synthase or histone H3 (Fig 2I, Appendix Fig S3N and O). Taken together, these results suggest that Nox4-dependent inhibition of PP1 specifically prolongs eIF2α phosphorylation in cardiac cells and MEFs, without affecting other PP1 targets.

### Nox4 binds to GADD34 to mediate localized PP1 inhibition

The targeting of PP1 to eIF2α involves GADD34 (Novoa *et al*, 2001), and recent work indicates that GADD34 interacts with both PP1 and eIF2α at the ER, thereby acting as a scaffold (Choy *et al*, 2015). To look for a more specific molecular mechanism underlying the inter-relationship between Nox4, GADD34, PP1 inhibition, and eIF2α, we first assessed the subcellular localization of Nox4 during ER stress. Using super-resolution 3D structural illumination microscopy (SIM), an anti-KDEL antibody that recognizes ER chaperones was used to visualize the ER (Fig 3A). After tunicamycin treatment, the induced Nox4 protein was sited in punctae along the tubular ER network, co-localizing with KDEL proteins. Confocal microscopy of cells co-transfected with Myc-tagged Nox4 and Flag-tagged GADD34 demonstrated co-localization of the two proteins at the ER, with similar results obtained for the endogenous proteins in tunicamycin-treated H9c2 cells (Fig EV1A–C).

We next assessed the relative distributions of Nox4, GADD34, PP1, and eIF2α in membrane and cytosolic fractions prepared from tunicamycin-treated H9c2 cells. Following treatment with tunicamycin, there was a marked enrichment of all 4 proteins in the membrane fraction (Fig 3B, Appendix Fig S4A), consistent with their recruitment and co-localization. To assess whether these proteins might be in a macromolecular complex, we subjected lysates of tunicamycin-treated H9c2 cells to sucrose gradient centrifugation, a method that can separate macromolecular complexes based on their size and shape (Meselson *et al*, 1957). This revealed that GADD34, PP1, eIF2α, and Nox4 co-eluted in fractions F12/F13, corresponding to fractions containing protein complexes of around 250 kDa, with a similar pattern also observed in Nox4-transfected cells (Fig 3C, Appendix Fig S4B and C). Immunoprecipitation of these fractions with an anti-GADD34 antibody revealed the presence of PP1 but, strikingly, also Nox4 (Fig 3D). The analysis of interaction between Nox4 and PP1 under similar conditions was hampered by the lack of suitable antibodies (data not shown). To validate the GADD34–Nox4 association, HEK cells were co-transfected with Nox4 and C-terminal Flag-tagged GADD34. Immunoprecipitation with an anti-Flag antibody pulled out both GADD34 and Nox4 (Fig 3E), supporting an interaction between the proteins. A non-tagged GADD34 construct was used as a negative control. We also performed co-immunoprecipitation studies with Myc-tagged constructs of full-length Nox4, the N-terminal transmembrane domain (Nox4-TD) or the C-terminal cytosolic domain (Nox4-CD), co-transfected into HEK cells along with GADD34-Flag. Anti-Myc immunoprecipitation revealed that both full-length Nox4 and Nox4-TD but not Nox4-CD associated with

    

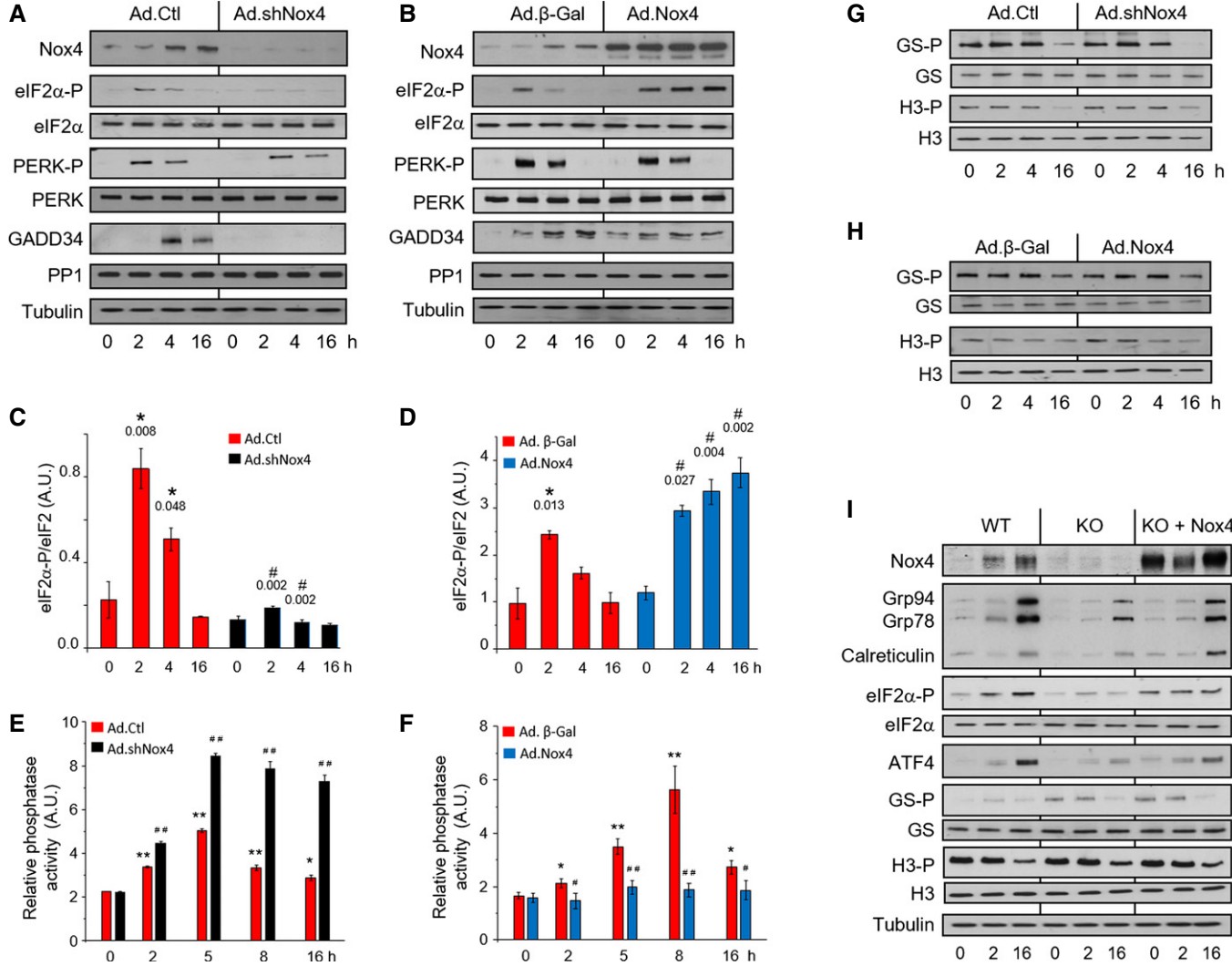

**Figure 2. Nox4 selectively inhibits PP1 activity and prolongs eIF2α phosphorylation.**

A   The knockdown of endogenous Nox4 resulted in a substantial inhibition of tunicamycin-induced eIF2α phosphorylation in H9c2 cells, with no change in phospho-Thr980-PERK (PERK-P) levels. GADD34 levels were significantly decreased after Nox4 knockdown, while there was no change in PP1 protein levels.

B   Overexpression of Nox4 in H9c2 cells caused prolongation of tunicamycin-induced eIF2α phosphorylation, with minimal change in phospho-PERK levels.

C, D   Mean levels of phosphorylated eIF2α relative to total eIF2α protein in tunicamycin-treated cells after Nox4 knockdown or overexpression, respectively. n = 3/group. *, significant compared to baseline; #, significant comparing Nox4 knockdown (Ad.shNox4) or overexpression (Ad.Nox4) versus corresponding controls (Ad.Ctl or Ad.β-Gal, respectively).

E, F   Effect of Nox4 knockdown or overexpression, respectively, on okadaic acid-resistant Ser/Thr phosphatase activity in membrane fractions of tunicamycin-treated H9c2 cells. n = 4/group. *P < 0.05, **P < 0.01 cf. baseline; #P < 0.05, ##P < 0.01 comparing Nox4 knockdown (Ad.shNox4) or overexpression (Ad.Nox4) versus corresponding controls (Ad.Ctl or Ad.β-Gal, respectively).

G, H   Nox4 knockdown or overexpression, respectively, had no effect on the phosphorylation of glycogen synthase at Ser641 (GS-P) or histone H3 at Ser57 (H3-P) in H9c2 cells.

I   Nox4$^{-/-}$ MEF cells (KO) showed blunted tunicamycin-induced increases in levels of phospho-eIF2α, ATF4, and ER chaperones as compared to wild-type (WT) MEFs, a response that was rescued by reintroduction of Nox4 (KO + Nox4). The latter had no effect on GS-P or H3-P levels.

Data information: All blots are representative of at least 3 independent experiments. Data are presented as mean ± SEM. Comparisons were made by ANOVA and P < 0.05 was considered significant. Values above bar graphs denote the level of significance. Mean data from quantification of immunoblots are shown in Supplementary Appendix Fig S3.

GADD34 (Fig 3F), suggesting that the Nox4 N-terminal domain is involved in GADD34 binding.

To directly assess the subcellular localization of Nox4-mediated ROS production during ER stress in living cells, we used a new multiparametric approach (Ermakova *et al*, 2014) to enable simultaneous imaging of $H_2O_2$ in the ER and cytosol. We studied

tunicamycin-stimulated Nox4$^{-/-}$ MEFs, WT MEFs, and Nox4$^{-/-}$ MEFs transfected with Nox4 (Appendix Fig S4D and E). WT MEFs showed a significant increase in ER ROS but not cytosolic ROS after tunicamycin treatment (Figs 3G and EV1D and E). The increase in ER ROS was lost in Nox4$^{-/-}$ MEFs but was restored in cells that had been transfected with Nox4. Exposure to extracellular $H_2O_2$ was

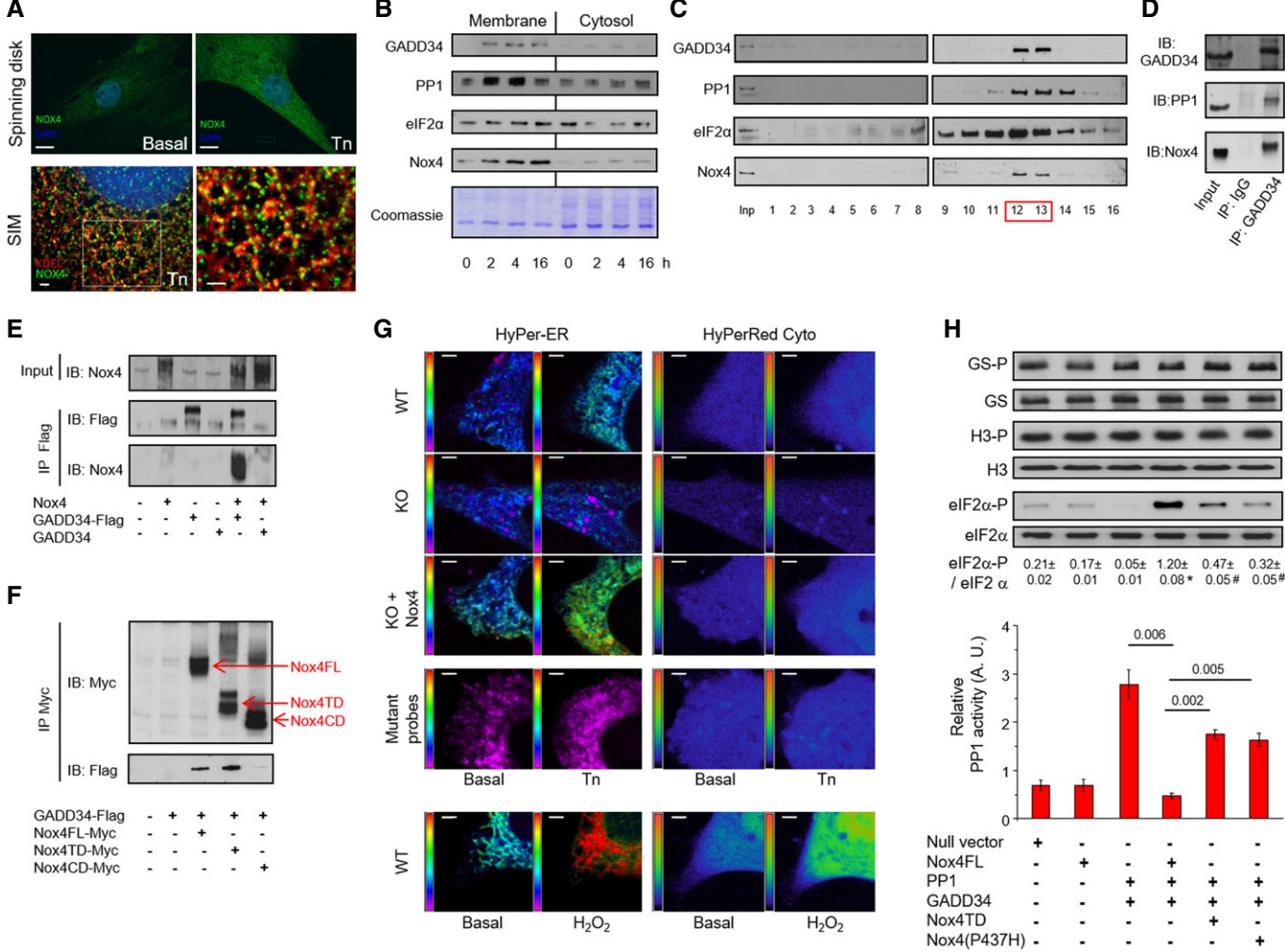

**Figure 3.  Nox4 binds to GADD34 to mediate spatially localized PP1 inhibition and enhance eIF2α phosphorylation.**

A   Subcellular localization of Nox4. Tunicamycin (Tn 2 μg/ml, 6 h) increased Nox4 levels in H9c2 cells as assessed by spinning disk confocal microscopy (scale bar, 10 μm). 3D SIM images (scale bars, 2 μm) showed localization of Nox4 (green) to the ER, which was labeled with an anti-KDEL antibody (red). At higher magnification (right), yellow dots denote co-localization of Nox4 and KDEL signals. Cell nuclei were stained with DAPI (blue). 1 Z slice from 3D stack is shown.

B   Progressive enrichment of GADD34, PP1, eIF2α, and Nox4 in membrane fractions of tunicamycin-treated H9c2 cells.

C   After sucrose gradient fractionation of lysates of tunicamycin-treated H9c2 cells, GADD34, PP1, eIF2α and Nox4 co-eluted in fractions 12 and 13 (F12, F13).

D   Immunoprecipitation (IP) of pooled fractions 12/13 with an anti-GADD34 antibody revealed the presence of both PP1 and Nox4.

E   The association of Nox4 with GADD34 was validated in HEK293 cells co-transfected with Nox4 and either Flag-tagged or non-tagged GADD34, followed by IP with an anti-Flag antibody.

F   Co-transfection of HEK293 cells with GADD34-Flag and different myc-tagged Nox4 constructs, followed by IP with an anti-myc antibody. GADD34 binds to full-length Nox4 (FL) and the Nox4 transmembrane domain (TD), but not the C-terminal domain (CD).

G   Representative pseudocolor images of simultaneous ER and cytosolic ROS measurement with HyPer-ER and HyPer-Red Cyto, respectively, in tunicamycin-treated MEF cells. Redox-insensitive mutant probes were used as negative controls and to exclude pH changes. Extracellular $H_2O_2$ (200 nM) was added as a positive control. The pseudocolor scale is shown along the left vertical edge of each image. KO = Nox4$^{-/-}$. Scale bars, 2 μm.

H   Transfection of HEK293 cells with PP1 and GADD34 increased PP1 activity (bar graph) and decreased phospho-eIF2α levels (immunoblots). Co-transfection of full-length Nox4 reduced PP1 activity and increased phospho-eIF2α levels (captions at bottom refer both to the bar graphs and immunoblots). These effects were abrogated when either Nox4 P437H or the Nox4 transmembrane domain (TD) was transfected. Nox4 did not affect phosphorylation of glycogen synthase (GS-P) or histone H3 (H3-P). Data are presented as mean ± SEM.

Data information: All experiments were performed with *n* = 3/group. Values below the immunoblots are mean ± SEM levels for phospho-eIF2α/total-eIF2α. *$P < 0.05$ comparing third and fourth lanes; #$P < 0.05$ compared to lane 4. All comparisons were made by Student's *t*-test, with $P < 0.05$ considered significant; levels of significance for comparisons of PP1 activities are shown above the bar columns. See also Fig EV1 and Appendix Fig S4.

used as a positive control and mutant redox-insensitive probes as negative controls. These results indicate that Nox4-dependent ROS generation occurs locally at the level of the ER in cells subjected to protein unfolding stress.

To confirm the role of Nox4 enzymatic activity in the modulation of PP1 activity and eIF2α phosphorylation, we studied HEK cells co-transfected with PP1 and GADD34 along with either full-length Nox4, Nox4-TD, or a mutant Nox4 construct with a single proline

to histidine amino acid substitution at residue 437 in the NADPH-binding domain, P437H-Nox4 (Dinauer *et al*, 1989). Both P437H-Nox4 and Nox4-TD, which lacks NADPH and FAD binding sites (Nisimoto *et al*, 2010), show a marked reduction in ROS production compared to full-length Nox4 (Fig EV1F). Co-transfection of PP1 and GADD34 alone resulted in significantly increased PP1 activity and reduction in phospho-eIF2α levels (Fig 3H). When full-length Nox4 was also included, the elevation in PP1 activity was abolished and phospho-eIF2α levels increased substantially, but with no change in levels of phosphorylated glycogen synthase or histone H3. However, transfection of either P437H-Nox4 or Nox4-TD caused only minor changes in PP1 activity and eIF2α phosphorylation.

Overall, these results reveal a highly specific and spatially localized redox mechanism by which Nox4 inhibits GADD34-bound PP1 at the ER.

### Redox inhibition of PP1 activity involves oxidation of its metal center

The redox inhibition of PTPs such as PTP1B is an archetypal mechanism underlying regulation of protein kinase signaling (Lee *et al*, 1998) and involves the reversible oxidation of a catalytic cysteine residue (Cys215) in the enzyme active site (van Montfort *et al*, 2003; Salmeen *et al*, 2003). PP1 relies on a dinuclear metal center (rather than cysteine) for catalysis (Egloff *et al*, 1995; Goldberg *et al*, 1995) and is considered to be regulated largely through binding to specific targeting and inhibitor proteins, with little known about its physiologic redox regulation (Heijman *et al*, 2013). We found that recombinant PP1 was concentration-dependently inactivated by $H_2O_2$ ($IC_{50}$ ~67 µM, Fig EV2A). Thiol reductants such as dithiothreitol (DTT), glutathione, and cysteine did not restore PP1 activity after removal of excess $H_2O_2$ by catalase treatment (Fig 4A). However, a one-electron reductant, ascorbate, efficiently reversed PP1 inactivation (Fig 4B). Electron paramagnetic resonance spectroscopy (EPR) demonstrated the formation of ascorbyl free radicals during incubation of oxidized PP1 with ascorbate (Fig 4C), confirming its reducing activity (Du *et al*, 2012). These results led us to consider the possibility that oxidative inactivation of PP1 involves redox changes at the dinuclear metal center.

We pursued an X-ray crystallographic approach to evaluate the effects of $H_2O_2$ on PP1 structure. The PP1 catalytic domain displays a compact ellipsoidal structure with its dinuclear metal center situated at the base of a shallow groove on the molecular surface (Egloff *et al*, 1995; Goldberg *et al*, 1995) (Fig EV2B). The exact metal center composition under physiologic conditions is not known. Recombinant PP1 is typically expressed in the presence of manganese salts, thus leading to a mostly Mn-Mn center, but a previous report suggested a mixed Mn-Fe center (Egloff *et al*, 1995). As Mn(II) and Fe(II) have similar coordination preferences, they could share the same site in PP1. Our metal analysis on PP1 crystals detected both Mn and Fe ions (Mn:Fe ratio ~1:0.175), with anomalous difference electron density maps revealing the presence of Mn at both coordination sites (Fig EV2C and D). Soaking of PP1 crystals with $H_2O_2$ did not induce major structural rearrangements and the metal centers remained pseudo-octahedrally coordinated with a bound phosphate moiety. The bridging µ-OH⁻ group is opposite the scissile P-O bond, suitably positioned for nucleophilic

attack on the phosphorus center with the assistance of His125 (Fig EV2D). While $H_2O_2$ treatment did not alter the general structural features of the active site or phosphate binding, we observed a contraction of the average metal coordination sphere by 0.12 Å compared to ascorbate-treated crystals (Fig 4D, Appendix Table S1). A transition from Mn(II) to Mn(III) or from Fe(II) to Fe(III) upon $H_2O_2$-dependent metal oxidation would shrink the ionic radius of the metal by up to 0.14 Å (Cotton *et al*, 1999), thereby shortening the bond distances between the metal and coordinating residues and increasing the energy barrier for phosphate hydrolysis. Interestingly, quantum chemical calculations predict a 0.15 Å shrinkage of the coordination sphere upon oxidation of an Mn(II) dinuclear center (Zhang *et al*, 2013; Appendix Table S1, Fig EV2E), supporting our observation of $H_2O_2$-dependent metal oxidation.

In our crystallographic studies, two cysteine residues, C127 and C273, were often observed oxidized to a sulfenic derivative in electron density maps (Fig EV2F). C127 and C273 are peripheral to the metal center and unlikely to affect enzyme activity, and we found that a PP1 C127/273S double variant exhibited the same catalytic activity as wild-type PP1 and responded identically to $H_2O_2$ treatment (Fig 4A). By contrast, PP1 variants in which the charge of amino acids involved in metal ion coordination was altered (i.e. N124D or D64N) showed enhanced $H_2O_2$-mediated inhibition (Fig EV2G and H), again pointing to the metal center as the site of oxidative inhibition. Finally, we used EPR at low temperature, a technique that detects transition metal ion complexes in proteins. Fe(II) is EPR-silent, in contrast to Fe(III), which is EPR-active (Cammack & Cooper, 1993). Both Mn(II) and Mn(III) are EPR-active, with no distinction between the species. PP1 showed an EPR spectrum composed of a main central six-line signal from 2.5 to 4.5 kG (Fig EV2I) consistent with $Mn^{2+}$ (or $Mn(H_2O)_6^{2+}$) (Reiter *et al*, 2002). Incubation of PP1 with $H_2O_2$ resulted in a small but consistent signal in the region of 1.5–2 kG, typical for Fe(III) and providing further evidence for the oxidation of Fe(II).

To provide evidence in support of the above mechanism in cells, we studied the effect of the N124D and D64N PP1 variants on eIF2α phosphorylation in HEK cells co-transfected with GADD34 and Nox4. Consistent with the enhanced $H_2O_2$-mediated inhibition of PP1 activity observed in these variants *in vitro*, PP1 N124D and D64N were both associated with higher phospho-eIF2α levels in the presence of Nox4 than seen with native PP1 (Fig 4E). We also studied the effect of ascorbate treatment on Nox4-modulated responses in tunicamycin-treated H9c2 cells. In control cells, ascorbate enhanced the increase in phosphatase activity observed after tunicamycin treatment (Fig 4F). The reduction in PP1 activity and increase in eIF2α phosphorylation observed in Nox4-overexpressing cells were both reversed by ascorbate, whereas ascorbate had no significant effect in Nox4-depleted cells (Figs 4F and EV2J).

Taken together, these results suggest that Nox4-dependent inhibition of PP1 involves oxidation of its metal coordination sites and a reduction in catalytic efficiency.

### Nox4 enhances cell survival during ER stress

To examine the functional consequences of Nox4-modulated PP1/eIF2α signaling, we studied the survival of tunicamycin-treated

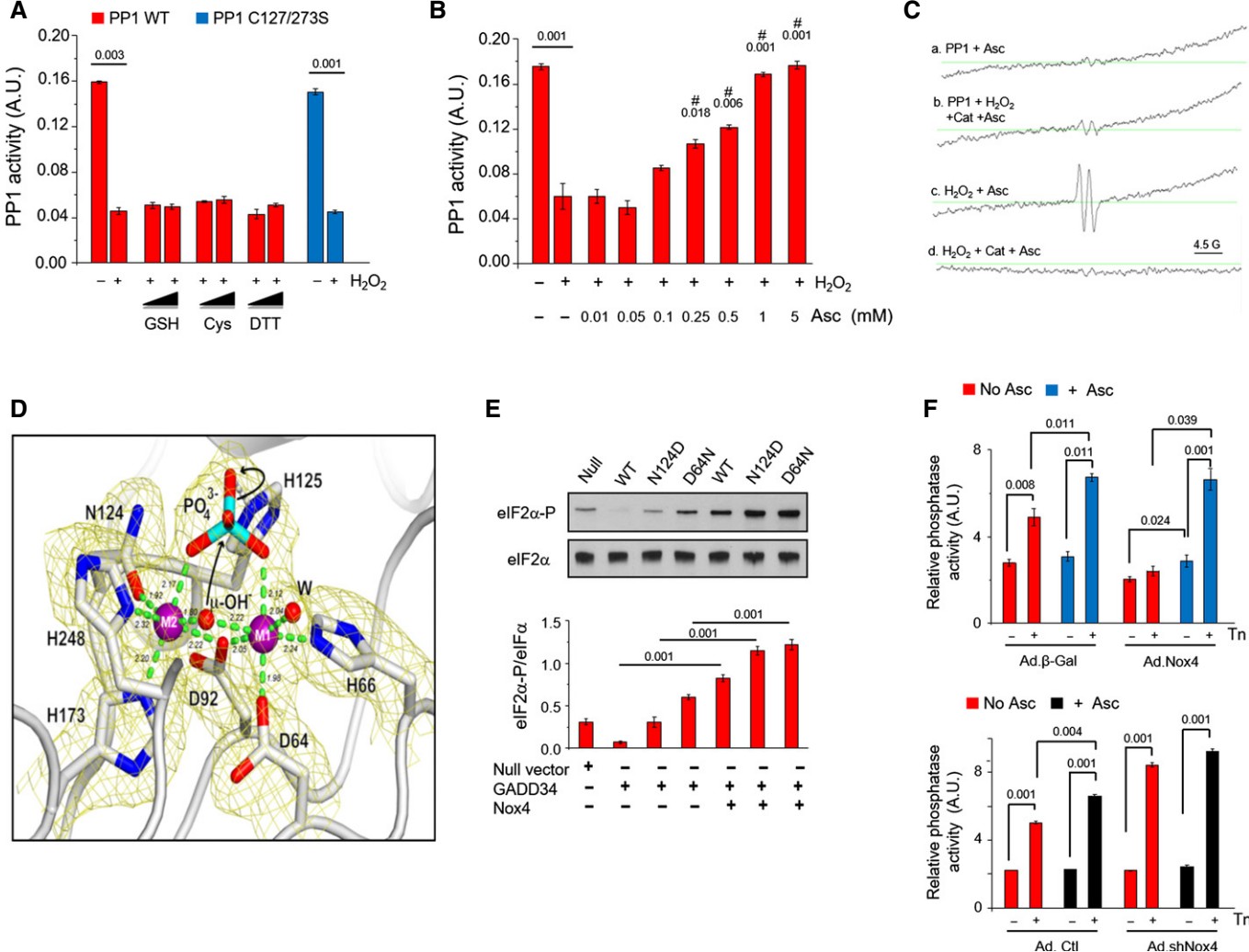

**Figure 4.   Redox inhibition of PP1 involves metal center oxidation.**

A   Recombinant PP1 was inhibited by $H_2O_2$ (0.2 mM) and activity was not restored by glutathione (GSH), cysteine (Cys), or dithiothreitol (DTT). A Cys127Ser/Cys273Ser PP1 mutant was inhibited by $H_2O_2$ similarly to wild-type PP1. Values above bars denote level of significance for the inhibitory effect of $H_2O_2$.

B   Ascorbate (Asc) dose-dependently restored PP1 activity. #, significant effect of Asc compared to $H_2O_2$ alone.

C   EPR spectra of PP1 incubated with ascorbate (1 mM) alone (a) or PP1 exposed to $H_2O_2$ followed by catalase treatment, then incubation with ascorbate (b). (b) shows a typical spectrum for the ascorbyl radical (hyperfine splitting constant, $a_H$ = 1.8 G), similar to the positive control obtained by exposing ascorbate to $H_2O_2$ (c). (d) shows that no ascorbyl radical is detected if $H_2O_2$ is degraded by catalase in the absence of PP1, prior to ascorbate addition.

D   Cartoon representation of the active site of $H_2O_2$-treated PP1 as in Fig EV2D. $2mF_o$-$DF_c$ electron density map at the 2.2-Å resolution is shown in yellow at the 1.1σ level. $H_2O_2$ treatment causes an overall shrinkage of the PP1 coordination sphere by 0.12 Å compared to ascorbate-treated crystals consistent with the oxidation of the dinuclear center. This increases the energy barrier for the catalytic steps involving μ-$OH^-$ attack on the phosphorous center of the bridging phosphate and rupture of the P-O scissile bond with the assistance of H125 (black arrows). Reported coordination distances in Å are averaged over the two PP1 molecules in the crystallographic asymmetric unit. See also Appendix Table S1.

E   Effect of Nox4 on GADD34/PP1-mediated eIF2α dephosphorylation in transfected HEK293 cells. Nox4 increased eIF2α phosphorylation in cells transfected with GADD34 and WT PP1, and resulted in even higher phospho-eIF2α levels in cells transfected with N124D or D64N PP1 variants.

F   Effect of ascorbate (Asc, 0.5 mM) on phosphatase inhibition in tunicamycin-treated H9c2 cells with overexpression or knockdown of Nox4 (Ad.Nox4 and Ad.shNox4, respectively). In control cells, ascorbate enhanced tunicamycin-stimulated increases in phosphatase activity. Phosphatase activity was lower in Nox4-overexpressing than control cells but was normalized by ascorbate to the same level as in control cells. In Nox4 knockdown cells, tunicamycin-induced increases in phosphatase activity were enhanced and ascorbate had minimal additional effect.

Data information: All experiments were performed with $n$ = 3/group. Data are presented as mean ± SEM. Comparisons were made by Student's $t$-test or one-way ANOVA, with $P < 0.05$ considered significant. Values above bar graphs denote the level of significance. See also Fig EV2..

cardiac cells. Nox4 knockdown significantly reduced cell survival after tunicamycin treatment, accompanied by a decrease in ATF4 levels and an increase in levels of cleaved caspase-3, a marker of apoptotic cell death (Fig 5A and B, Appendix Fig S5). We used a

small molecule agent, guanabenz, which selectively inhibits GADD34-bound PP1 (Tsaytler *et al*, 2011), to specifically establish the contribution of the GADD34/PP1/eIF2α pathway to these effects. We reasoned that by mimicking the effects of Nox4 to specifically

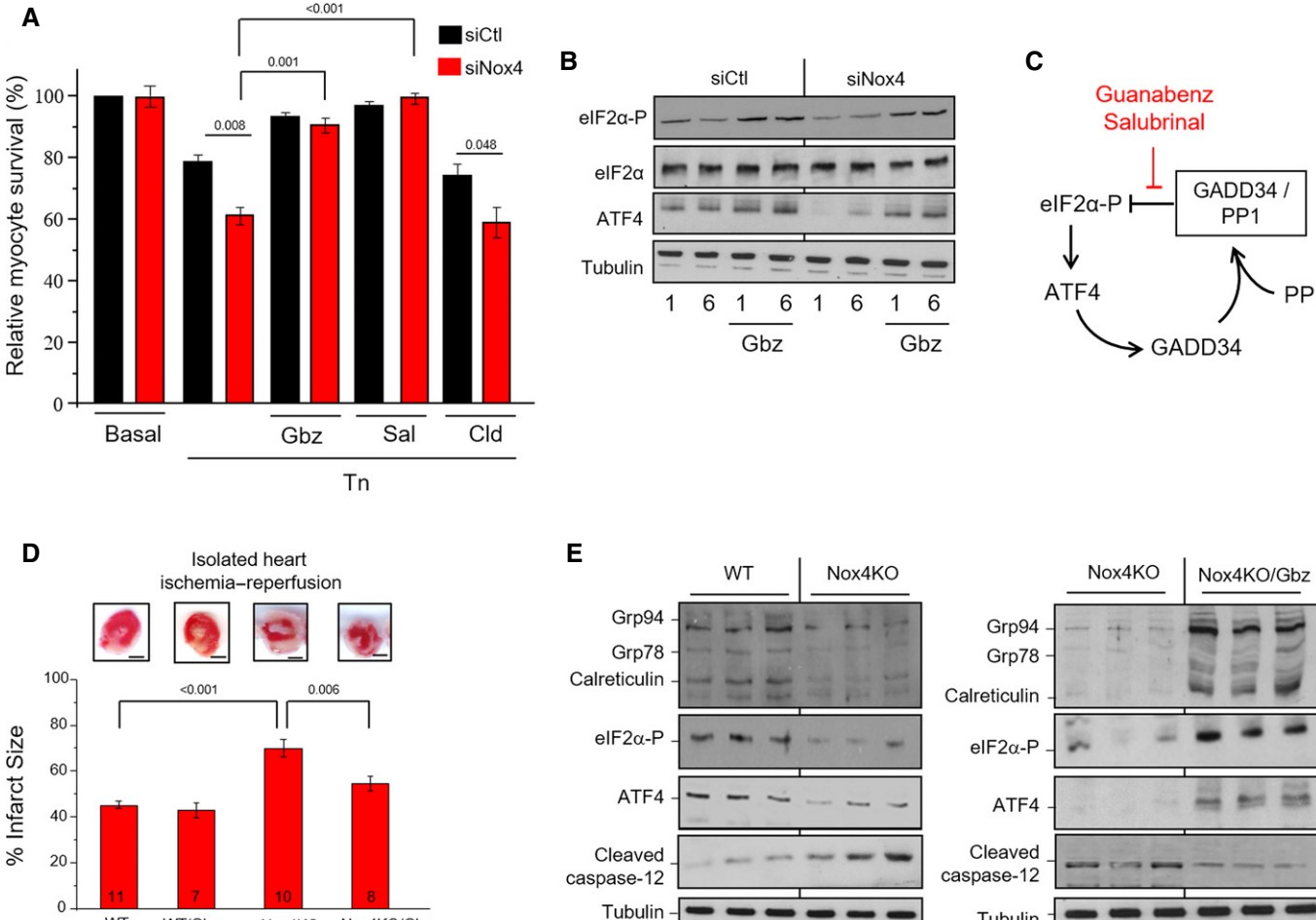

**Figure 5. Nox4 enhances cell survival and protects hearts against I/R injury through increase in eIF2α phosphorylation.**

A  H9c2 cells treated with tunicamycin (2 μg/ml, 12 h) showed significantly lower survival when endogenous Nox4 was silenced (siNox4) as compared to cells treated with a scrambled siRNA (siCtl). Cell survival was restored by treatment with either guanabenz (Gbz, 5 μM) or salubrinal (Sal, 50 μM) but was unaffected by clonidine (Cld, 5 μM). n = 3/group.

B  Nox4-depleted cells had lower levels of phospho-eIF2α and ATF4 than control cells, but these were restored in the presence of guanabenz (Gbz).

C  Schematic representation of the effect of the small molecule inhibitors, guanabenz and salubrinal, on the GADD34/PP1/eIF2α interaction.

D  Hearts from Nox4 knockout (KO) mice and WT controls were subjected to global ischemia followed by aerobic reperfusion (I/R). Infarct size assessed by triphenyltetrazolium chloride (TTC) staining was greater in Nox4 KO hearts compared to WT and was significantly reduced by guanabenz (Gbz). In the representative heart sections shown at the top, white denotes infarct area and red the viable area. Scale bars, 1 mm. Numbers of hearts are indicated within the bars.

E  Immunoblotting of heart homogenates after I/R showed lower levels of phospho-eIF2α, ATF4, and ER chaperones, and higher levels of cleaved caspase-12, in Nox4 KO compared to WT. Tubulin was used as a loading control. Treatment with guanabenz (Gbz) reversed these changes (blots shown to the right).

Data information: Data are presented as mean ± SEM. Comparisons were made by one-way ANOVA, with $P < 0.05$ considered significant. Values above bar graphs denote the level of significance. See also Appendix Fig S5.

inhibit GADD34-bound PP1, guanabenz should rescue the survival defect in Nox4-deficient cells (Fig 5C). As expected, guanabenz enhanced eIF2α phosphorylation in control cells at a late time-point when it would normally be declining and increased cell survival (Fig 5A and B, Appendix Fig S5). Strikingly, guanabenz also enhanced eIF2α phosphorylation and reduced cleaved caspase-3 levels in stressed Nox4-deficient cells, and increased survival to a level identical to control cells. Guanabenz has α2-adrenergic receptor agonist activity but an α2-agonist, clonidine, that does not alter PP1 activity (Tsaytler *et al*, 2011) had no effect on cell survival (Fig 5A). Furthermore, similar beneficial effects to guanabenz were observed with a structurally distinct inhibitor of GADD34/PP1 action, salubrinal (Boyce *et al*, 2005). Nox4-mediated inhibition of

GADD34/PP1 signaling therefore plays an essential role in sustaining cell survival during ER stress.

### Nox4 is protective against acute heart and kidney injury

We next investigated the role of Nox4-mediated regulation of GADD34/PP1 signaling in acute heart injury, a setting where the UPR is robustly activated (Thuerauf *et al*, 2006) and Nox4 expression also increases (Zhang *et al*, 2010; Matsushima *et al*, 2013). Hearts from Nox4 knockout (KO) mice and WT controls were subjected to ischemia–reperfusion and the extent of infarction was assessed. We found that Nox4 KO hearts developed significantly larger infarcts than WT, had lower levels of phospho-eIF2α, ATF4,

and KDEL proteins, and had higher levels of cleaved caspase-12 (Fig 5D–F), consistent with an increase in ER stress-associated cell death. Treatment with guanabenz substantially reduced infarct size in Nox4 KO hearts, increased the levels of phospho-eIF2α, ATF4, and KDEL proteins, and decreased cleaved caspase-12 levels. Guanabenz had no significant effect on infarct size in WT hearts.

To investigate whether similar protective effects are important in other organs *in vivo*, we studied a murine model of acute kidney injury (AKI) induced by intraperitoneal injection of tunicamycin, which involves ER stress-related injury (Zinszner *et al*, 1998). In this model, Nox4 KO mice developed substantially worse renal impairment than WT controls, with higher increases in blood urea levels (Fig 6A). The kidneys of Nox4 KO mice showed marked surface pallor, increased TUNEL-positive cells on histological examination, and higher levels of cleaved caspase 12 and cleaved poly-ADP ribose polymerase (PARP), two markers of apoptosis (Fig 6B–D). Most strikingly, survival was substantially lower in Nox4 KO compared to WT mice (Fig 6E). Pre-treatment

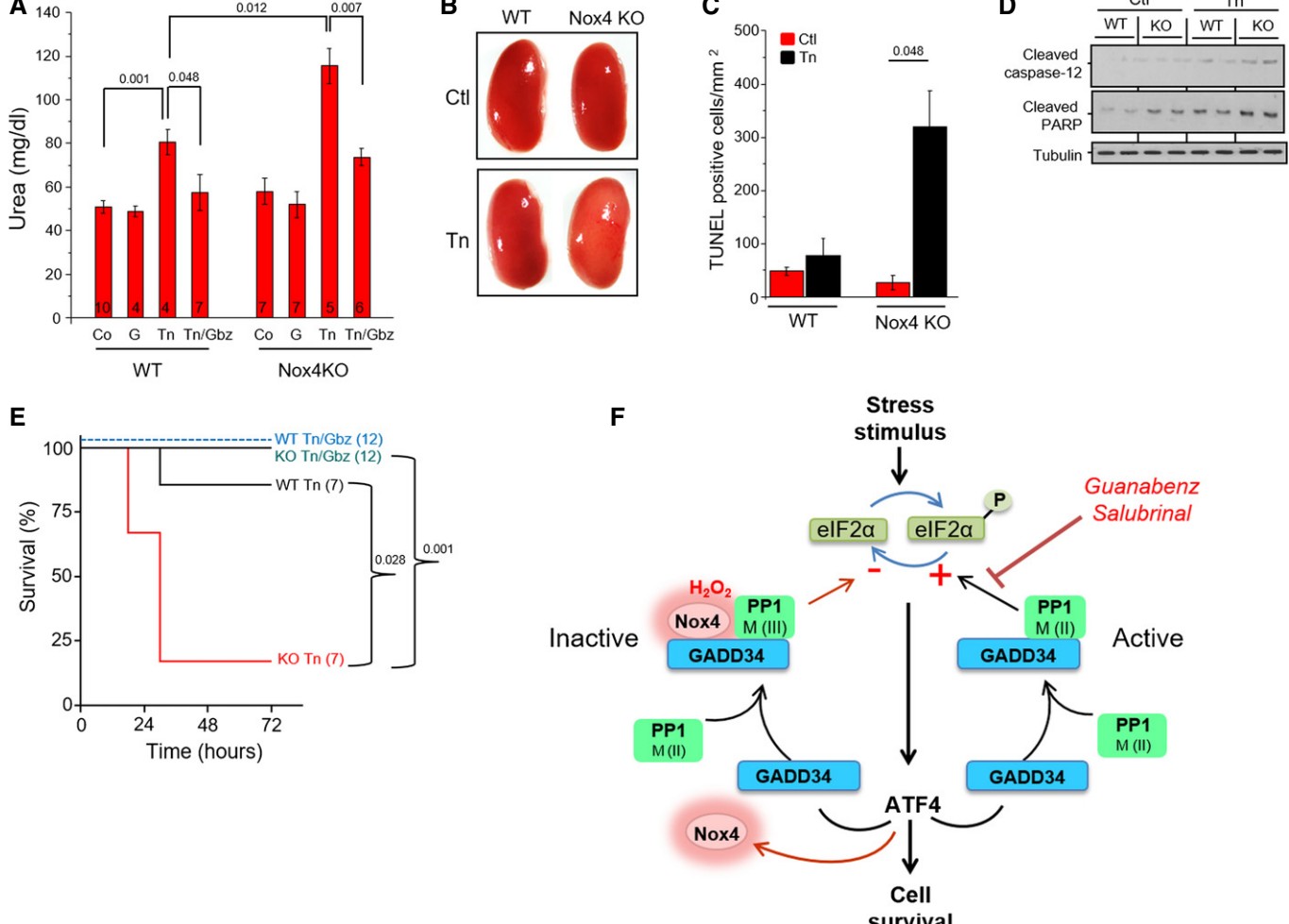

**Figure 6. Nox4 protects against acute kidney injury *in vivo*.**

A   Plasma urea levels were elevated to a greater extent in tunicamycin-treated Nox4 KO mice than WT. Co-treatment with guanabenz (Gbz) reduced urea levels in both groups. Numbers of animals are indicated within bars.

B   Forty-eight hours after systemic tunicamycin treatment, kidneys of Nox4 KO mice showed a marked surface pallor (bottom right).

C   TUNEL staining revealed a significantly higher number of apoptotic cells in tunicamycin-treated Nox4 KO mice. n = 4/group.

D   Immunoblotting of kidney homogenates showed significantly elevated cleaved caspase-12 and cleaved PARP levels in tunicamycin-treated Nox4 KO mice compared to WT.

E   Survival curves showed that a very high proportion of Nox4 KO mice died after AKI. Guanabenz (Gbz) treatment dramatically improved survival in tunicamycin-treated KO mice. Number of animals as indicated. Levels of significance by Kaplan–Meier analysis are reported to the right.

F   Schematic depicting the effect of Nox4-generated ROS on PP1 activity and the balance between eIF2α phosphorylation and dephosphorylation. Nox4 is upregulated by ATF4 and binds to GADD34. It inhibits GADD34-bound PP1 through the local generation of $H_2O_2$ and oxidation of the metal (M) center of the serine–threonine phosphatase. The consequent prolongation of eIF2α phosphorylation promotes cell survival in the face of acute protein unfolding stress. M = iron or manganese, which are oxidized from the M (II) to the M (III) species.

Data information: Data are presented as mean ± SEM. Comparisons in (A, C) were made by one-way ANOVA, with *P* < 0.05 considered significant. Values above bar graphs denote the level of significance.

with guanabenz substantially reduced the elevation in blood urea levels in both WT and Nox4KO animals and, remarkably, prevented death of the tunicamycin-treated Nox4KO mice (Fig 6A and E).

Collectively, these results indicate that Nox4-dependent regulation of GADD34/PP1/eIF2α signaling makes a crucial contribution to protection against acute heart and kidney injury.

## Discussion

The effectiveness of the ISR is manifest through several layers of regulation that allow for fine-tuning and integration with other physiologic pathways (Walter & Ron, 2011; Baird & Wek, 2012). In response to ER stress, the phosphorylation of eIF2α results in a reduction in synthesis of most ER proteins (and therefore a decrease in protein folding load) but is accompanied by the increased translation of a subset of proteins, notably ATF4. ATF4 induces numerous genes that may contribute to stress resistance (Harding et al, 2003), including ER chaperones which increase folding capacity (Luo et al, 2003; Ma & Hendershot, 2003). A second level of regulation that influences cell fate involves the set point for eIF2α phosphorylation which is regulated by the GADD34–PP1 complex, with GADD34 itself induced downstream of ATF4 and leading to eIF2α dephosphorylation (Kojima et al, 2003; Tsaytler et al, 2011). Here, we report another layer of control whereby GADD34-associated PP1 is inhibited by a dedicated ROS-generating protein, Nox4, such that the eIF2α-dependent adaptive response is boosted to promote cell survival (Fig 6F). We identify the specific signaling interactions that orchestrate this regulatory loop, including the findings that the level of Nox4 itself is further increased by ATF4 and that it binds to GADD34 to mediate a spatially confined redox inhibition of PP1, achieved through a novel metal center oxidation mechanism. This results in a bidirectional positive feedback loop between Nox4 and ATF4, mediated through altered eIF2α phosphorylation. Thus, on the one hand, Nox4 amplifies the increase in ATF4 that occurs during the ISR by boosting eIF2α phosphorylation while, on the other hand, ATF4 itself increases Nox4 levels.

Interestingly, ER stress-induced increases in eIF2α phosphorylation and ATF4 levels are modulated even by the Nox4 levels that are present basally (as revealed by experiments involving the silencing of Nox4). Nox4 levels then rise rapidly following exposure to tunicamycin or thapsigargin, a response that is at least in part mediated by the transcriptional effects of ATF4 but could also involve non-transcriptional mechanisms such as changes in protein stability, as reported recently in other contexts (Desai et al, 2014). Since Nox4 levels are increased by diverse stresses, including hypoxia, ischemia, cytokines, and mechanical forces (Lassègue et al, 2012), it is likely that the ATF4-dependent upregulation of Nox4 driven by protein unfolding stress may synergize with increases in Nox4 levels driven by other stress stimuli to fine-tune eIF2α phosphorylation and thereby provide a more integrated cellular response to ambient conditions.

While unregulated ROS production during cellular stress responses may promote cell death (Murphy, 2009; Han et al, 2013), specific redox signaling typically depends upon spatially confined ROS production and reversible atomic-level redox modifications in target proteins (D'Autréaux & Toledano, 2007). Such signaling

reconciles the apparent paradox of subtle, often adaptive responses versus indiscriminate ROS toxicity. The Nox4-regulated pathway identified here achieves signaling and context specificity through protein–protein interactions that precisely juxtapose ROS generator and target. The specific association of Nox4 with GADD34 may allow sufficient $H_2O_2$ accumulation in the immediate vicinity of GADD34-bound PP1 to inhibit just this subpopulation of PP1, and therefore eIF2α at the ER, without affecting more distant PP1 targets. Given previous reports suggesting compartmentation of Nox4-dependent signaling, although without elucidation of molecular details (Chen et al, 2008; Wu et al, 2010), our results suggest that specificity through protein–protein interaction might be a general paradigm for the orchestration of Nox4-regulated redox signaling.

The reversible thiol oxidation of PTPs or PTEN (phosphatase and tensin homolog) is a well-established amplification mechanism in growth factor signaling (Lee et al, 1998; Kwon et al, 2004). Elucidation of metal center oxidation as the atomic-level mechanism underlying redox modulation of serine–threonine PP1 activity establishes an entirely distinct mode of redox regulation. Oxidation of the Mn/Fe PP1 metal center with concomitant shrinkage of the coordination sphere is expected to increase the energy barrier for phosphate hydrolysis (Zhang et al, 2013), in contrast to PTP1B inhibition where oxidation of Cys215 in the active site leads to a conformational change that inhibits phosphate binding (van Montfort et al, 2003; Salmeen et al, 2003). The precise metal species involved in redox inhibition of PP1 in vivo remains unclear. The overlapping metal binding sites are histidine-rich, suggesting a thermodynamic preference for Fe(II) (Hider & Kong, 2013); histidine binds Fe(II) more tightly than Mn(II), with $logK_1$ values of 5.6 and 3.0, respectively (Martell & Smith, 1974). Indeed, histidine-rich metal centers in many mammalian enzymes (e.g. the 2-oxoglutarate-dependent dioxygenases and ribonucleotide reductase) bind Fe(II) under normal physiologic circumstances. Thus, although our studies on recombinant PP1 found lower levels of Fe than Mn, the more abundant species in vivo may be Fe. Interestingly, the serine–threonine phosphatase calcineurin (or PP2a), which contains an Fe(II)/Zn(II) active center, was suggested to undergo metal center oxidation and inactivation by ROS, although structural details or linkages to redox signaling were not investigated (Namgaladze et al, 2002). With regard to Nox4-dependent modulation of PP1, it is of interest that Nox4 generates predominantly $H_2O_2$ rather than superoxide (Lassègue et al, 2012) since the direct oxidation of Fe(II)/Mn(II) is favored with the former species (Halliwell & Gutteridge, 1999). Intriguingly, Nox4 was also reported to inhibit HIF prolyl-4-hydroxylases, which contain non-heme Fe(II) and may be susceptible to oxidative modulation (Gerald et al, 2004). Detailed atomic-level analysis of the inhibition mechanism for these enzymes may be instructive in establishing whether Nox4 as an $H_2O_2$-generating oxidase specifically regulates redox-sensitive metal ions in diverse biological targets.

The UPR has a central role in many human diseases (Hetz et al, 2013), including conditions with major morbidity and mortality such as acute myocardial infarction and AKI (Thuerauf et al, 2006; Kitamura, 2008). The activation of the eIF2α limb of the UPR plays a crucial role in adaptation to acute stresses through the fine-tuning of protein synthesis and ATF4-mediated responses. Using loss-of-function and small molecule inhibitor strategies, here we show that

Nox4-mediated inhibition of eIF2α dephosphorylation significantly reduces the extent of cardiac injury during I/R and has even more dramatic beneficial effects during AKI. In the setting of acute myocardial infarction, pre-treatment with guanabenz (which enhances eIF2α phosphorylation and increases ATF4 levels) largely corrected the deficit in Nox4 KO hearts but had no additional effect in WT hearts. This suggests that the Nox4-dependent regulation of eIF2α/ATF4 signaling cannot be further enhanced in normal hearts during acute I/R and that other mechanisms also contribute to cardiac injury. However, in the setting of AKI, guanabenz improved outcome in both Nox4 KO and WT mice, suggesting that protective eIF2α/ATF4 signaling can be further boosted during AKI in normal mice with competent Nox4 regulation. Our results provide proof of concept for the existence and importance of Nox4-dependent regulation of eIF2α/ATF4 signaling during acute heart and kidney injury, and suggest that targeting the eIF2α pathway merits detailed assessment in these conditions. However, the consequences of redox-mediated regulation of eIF2α dephosphorylation in more chronic situations could be different (Moreno *et al*, 2012) and require further study.

In conclusion, Nox4-regulated inhibition of PP1 specifically delays eIF2α dephosphorylation during acute stress to boost cytoprotective signaling. Our work uncovers a novel redox signaling pathway orchestrated through protein–protein interaction between GADD34 and Nox4, and involving a hitherto unrecognized metal center oxidation mechanism, which regulates the ISR. This pathway exerts robust protective effects against acute cardiac and kidney injury.

# Materials and Methods

Unless otherwise indicated, all chemicals were purchased from Sigma-Aldrich or Calbiochem and were of analytical or higher purity grade. Detailed methods are provided in the Appendix.

### Cells and transfections

Primary cultures of neonatal rat cardiomyocytes were prepared using standard methods (Zhang *et al*, 2010). Rat H9c2 cardiomyoblasts, and HEK293 and U2OS cells were from ATCC. MEFs were prepared from 13.5-day-old embryos of Nox4$^{-/-}$ and littermate WT mice, and immortalized with SV40 large T antigen. Cell transfections with plasmids or siRNA were performed using Lipofectamine or FuGENE reagent (Promega). Plasmid sources were as follows: ATF4 (Addgene #24874), GADD34-Flag Mouse MyD116.P-FLAG.CMV2 (Addgene # 21834), GADD34 Mouse MyD116.delC.pBABEpu (Addgene # 21835), and pBABE-puro SV40 LT (Addgene # 13970). siRNA sequences are provided in the Appendix. Cell viability was assessed using an MTT assay.

### Nox4 constructs

Adenoviruses expressing Nox4 (Ad.Nox4), β-galactosidase (Ad.βGal), a short hairpin sequence targeted against Nox4 (Ad.shRNA.Nox4), or a short hairpin sequence targeted against GFP as control (Ad.Ctl) were infected at a multiplicity of infection (MOI) of 20, and cells were used 48 h later (Peterson *et al*, 2009; Zhang

*et al*, 2010). Nox4 deletion constructs were generated from a pcDNA3.1-full-length Nox4-myc plasmid (Anilkumar *et al*, 2008). The Nox4 P437H mutant was subcloned from the pcDNA3.1-full-length Nox4. This mutation inhibits NADPH binding and ROS generation in NADPH oxidases (Dinauer *et al*, 1989; Debeurme *et al*, 2010).

### Confocal and super-resolution microscopy

Confocal imaging was performed on a Nikon Ti-Eclipse microscope equipped with a Yokagawa CSU-X1-M2 spinning disk unit. 3D Structural illumination microscopy (SIM) was performed on a Nikon SIM system equipped with a 100× oil immersion objective. SIM image stacks were acquired with a z-distance of 100 nm and with 15 raw images per plane, five phases, three angles. Raw data were reconstructed using the SIM module in the NIS Elements software (NIKON). Images displayed are reconstructions of one z plane.

### ROS levels and ROS imaging

Cellular ROS levels and Nox activity were measured using HPLC-based detection of the oxidation products of dihydroethidium (DHE, Invitrogen), that is, 2- hydroxyethidium (EOH) and ethidium (E), as previously described (Laurindo *et al*, 2008). Nox activity was measured as NADPH-stimulated ROS generation in membrane fractions. Quantification of EOH and E was performed by comparison of peak signal between the samples and standard solutions under identical chromatographic conditions. Results are expressed as ratios of EOH and E generated per DHE consumed (initial DHE concentration minus remaining DHE; EOH/DHE and E/DHE).

Simultaneous imaging of ER and cytosolic ROS was undertaken in cells co-transfected with ER-targeted HyPer and cytosolic HyPer-Red (Belousov *et al*, 2006; Ermakova *et al*, 2014). The Cys199 residue in HyPer proteins is highly susceptible to oxidation by $H_2O_2$, resulting in a change in protein conformation and in fluorescence that can be visualized by imaging. HyPer emits green fluorescence upon oxidation whereas HyPer-Red emits red fluorescence. The respective C199S mutant probes for HyPer-ER and HyPer-Red, which are ROS-insensitive, were used as negative controls and to exclude changes in pH. Transfected cells were kept in phenol red-free medium supplemented with 2 mM glutamine and antibiotics for 48 h before treatment with tunicamycin (2 μg/ml for 4 h) or control vehicle. Imaging was performed at 37°C/5% $CO_2$ on a Nikon Ti-E microscope equipped with a Yokogawa CSU-X1 spinning disk confocal unit, an Andor Neo sCMOS camera, and a 60× objective. HyPer-ER fluorescence emission was monitored at 525/50 nm following excitation at 405 nm and 488 nm, and the ratio of fluorescence intensity was quantified. HyPer-Red fluorescence emission was monitored at 647/75 nm following excitation at 560/40 nm. Extracellular $H_2O_2$ (200 nM) was added as a positive control and the HyPer-ER and HyPer-Red signals acquired simultaneously. NIS Elements v.4.0 software (Nikon) was used for image analysis. Images were background-subtracted and thresholded. Changes in HyPer-ER fluorescence ratio (ΔR) or HyPer-Red fluorescence intensity (ΔF) between the indicated time-points or treatments were quantified. The resulting images were displayed in pseudocolor.

## Sucrose gradient fractionation

H9c2 cells were scraped, transferred into tubes, and centrifuged at 2,800 $g$ at 4°C for 5 min. The cell pellet was resuspended in 250 μl lysis buffer (50 mM Tris–HCl pH 7.2, 150 mM NaCl, 2 mM EDTA, 0.5% Triton X-100 containing protease cocktail and a proteasome inhibitor Mg132 2 μg/ml). Cell lysates were laid at the top of a sucrose gradient (10%, 20%, 40%, and 60%, top to bottom) prepared in 50 mM HEPES buffer pH 7.5, containing 100 mM KCl, 2 mM $MgCl_2$, 1 mM EGTA, and 1 mM EDTA. Samples were centrifuged at 35,000 $g$ (4°C, 18 h). Fractions 1–16 (F1–F16) were collected from the base of the column. Each fraction was split into two 200 μl aliquots, one for immunoblotting and the other for immunoprecipitation experiments. As a control for density gradient separation, a mix of proteins (Gel filtration molecular weight markers, Sigma-Aldrich) was added to the top of the sucrose gradient in a separate tube and centrifuged. The fractions obtained were submitted to SDS–PAGE, and proteins were stained with Coomassie Blue.

## Other cell assays

Immunoblotting, immunoprecipitation, real-time RT–PCR, cloning, and chromatin immunoprecipitation (ChIP) assay were performed using standard protocols. Antibody sources are listed in the Appendix, and primer sequences are listed in Appendix Table S2.

## EPR

For detection of ascorbyl radical, EPR spectra were recorded at room temperature on a Magnettech Miniscope MS2000 spectrometer. Instrument conditions were as follows: microwave power 50 mW, modulation amplitude 1 Gauss (G), and scan time 328 ms, with a gain of $9 \times 10^2$. These were calibrated with 4-hydroxy-2,2,6,6-tetramethyl-1-piperidinyloxy (Tempol). All spectra were the accumulation of 4 scans and were recorded 5 min after addition of $H_2O_2$. The reaction was carried out in 0.1 mM Tris buffer, pH 7.0, 37°C under the different conditions, and was transferred to a 50-μl flat cell immediately after addition of ascorbate. The two-line spectrum was consistent with an ascorbyl radical with a hyperfine splitting constant ($a_H$ = 1.8 G) (Monteiro *et al*, 2007), as generated using the positive control ascorbate and $H_2O_2$.

Assessment of transition metal redox state was performed by EPR at low temperature (4 K) on a Bruker EMX300 spectrometer with a 3-mm cavity and a helium cooling system (Cammack & Cooper, 1993; Ubbink *et al*, 2002). Purified PP1 (5 mg/ml) was studied at baseline and after treatment with $H_2O_2$ (1 mM) in Tris–HCl buffer pH 7.2 at 37°C. The reaction mixture was transferred to a flat cell and frozen in liquid nitrogen. Spectrometer conditions were as follows: temperature, 4 K; microwave frequency, 9.66 GHz; modulation amplitude, 2 G at 100 kHz; and microwave power, 20 mW.

## PP1 protein production

Untagged γ-isoform of the catalytic subunit of human PP1 and its variants were overexpressed and purified using published protocols (Alessi *et al*, 1993; Barford & Keller, 1994). Transformed *E. coli* DH5α cells were grown in Luria–Bertani (LB) medium supplemented with 2 mM $MnCl_2$ and 100 μg/ml ampicillin at 30°C until $OD_{600}$ reached ~0.25. Protein expression was induced with 0.5 mM IPTG. Cells were harvested by centrifugation and resuspended in buffer A (50 mM imidazole, 0.5 mM EDTA, 0.5 mM EGTA, 100 mM NaCl, 10% glycerol, 2 mM β-mercaptoethanol, 2 mM $MnCl_2$, pH 7.5) supplemented with Complete EDTA-free protease inhibitor cocktail, lysozyme (0.01 mg/ml), and DNase (0.05 mg/ml). After cell lysis, insoluble material was sedimented by centrifugation and the supernatant filtered using 0.22-μm filter prior to loading on a 5-ml heparin column equilibrated with buffer A. PP1 was eluted using a 100-ml gradient to 50% buffer A supplemented with 1 M NaCl. Fractions were analyzed on a 12% SDS–PAGE gel and those containing PP1 were pooled and diluted 10-fold with buffer C (50 mM imidazole, 0.5 mM EDTA, 0.5 mM EGTA, 10% glycerol, 5 mM β-mercaptoethanol, 2 mM $MnCl_2$, pH 7.2) for injection in a HiTrapQ HP (GE Healthcare) column. PP1 was eluted using a gradient to 40% buffer C supplemented with 1 M NaCl. PP1 was further purified by size-exclusion chromatography (SEC) using a Superdex 75 16/60 (GE Healthcare) column equilibrated with SEC buffer (50 mM imidazole, 0.5 mM EDTA, 0.5 mM EGTA, 300 mM NaCl, 10% glycerol, 5 mM β-mercaptoethanol, 2 mM $MnCl_2$, pH 7.5) for downstream applications. PP1 mutations (PP1 N124D and PP1 D64N) were introduced using the Q5 Site-Directed Mutagenesis kit (New England Biolabs). All constructs were verified by sequencing. Expression and purification of PP1 variants were carried out as for wild-type PP1.

## X-ray crystallography and metal analysis

PP1 crystals belonging to the space group $P2_1$ grew at 18°C using the vapor diffusion technique in either 20% PEG2000 MME, 200 mM NaCl, 0.1 M Tris–HCl pH 9.0 or 7–12% PEG3350, 0.1 M Bicine, pH 9.0. To ensure the reduced state of the PP1 dinuclear center, PP1 crystals were soaked with a reservoir solution enriched with 25 mM sodium ascorbate. Crystals were cryoprotected by soaking them in their respective reservoir solutions supplemented with either 20% glycerol or 25% 2-methyl-2,4-pentanediol (MPD) for a few seconds. PP1 oxidation was carried out by soaking PP1 crystals in 50 mM $H_2O_2$. Crystallographic data collections were performed at the Diamond Light Source (Oxford, UK) using beamlines I04-1 and I03. Model coordinates have been deposited with the Protein Data Bank with codes 4UT2 and 4UT3 for ascorbate-treated and hydrogen peroxide-treated PP1, respectively. The structure of PP1 was solved by the molecular replacement technique using the program MOLREP (Vagin & Teplyakov, 1997) starting from the coordinates of rat PP1 (PDB code 2O8A) as search model (Hurley *et al*, 2007). Full details of model refinement, without restraints for metal–ligand distances, are provided in the Appendix. A summary of data collection and refinement statistics is shown in Appendix Table S3.

Metal content of PP1 crystals was assessed by collecting a fluorescence spectrum using an X-ray excitation energy of 18 keV. Fitting was performed with the PyMCA package (Solé *et al*, 2007).

## PP1 reactions and activity

The activity of recombinant PP1c *in vitro* was measured using a Malachite Green assay (Millipore) in which the hydrolysis of a phospho-threonine peptide (Lys-Arg-phosphoThr-Iso-Arg-Arg, K-R-pT-I-R-R) was quantified spectrophotometrically at 620 nm

using a Nanodrop spectrophotometer. Buffers were pre-treated with Chelex-100 to remove transition metal ion contamination. Reaction mixtures containing PP1 (17 µg/ml) were treated with different concentrations of $H_2O_2$ for 15 min at 37°C and then further treated with catalase (30 U/ml) for 10 min at room temperature to remove residual $H_2O_2$. In experiments with DTT, cysteine, and GSH, these were added at a concentration of 0.5 and 1 mM each for 30 min. Ascorbate was added for 30 min at the concentrations indicated in the figures. For cellular activity of PP1, cell membrane extracts (Hubbard *et al*, 1990) were incubated with phosphopeptide substrate (0.1 mM) in the presence or absence of okadaic acid (10 nM), which does not inhibit PP1 at this concentration (Ishihara *et al*, 1989), and then, phosphatase activity was estimated as described above. For each sample, incubation without the phospho-peptide substrate was used as a blank. PP1 activity was taken as the okadaic acid-resistant fraction. Calyculin A (60 nM) (which inhibits both PP1 and PP2a) (Ishihara *et al*, 1989) was used as a control to confirm total PP activity. In some experiments, ascorbate (0.5 mM) was added to cells for 30 min before cell lysis.

### Animal studies

All procedures were performed in compliance with the UK Home Office "Guidance on the Operation of the Animals" (Scientific Proce-dures) Act, 1986. Nox4$^{-/-}$ and matched WT littermate mice on a C57BL6 background were used (Zhang *et al*, 2010).

Heart I/R injury was studied in isolated Langendorff-perfused hearts subjected to 25-min global ischemia and 100-min reperfusion after a 20-min equilibration period. Hearts were weighed, frozen, and cut into 1-mm-thick slices. Infarct area was calculated using 2,3,5-triphenyl-tetrazolium-chloride (TTC, 1% in phosphate buffer), which stains viable tissue red. Sections were then immersed in formalin and scanned. Infarct area was calculated as a proportion of the total left ventricular area using Image J Software. Some animals were treated with guanabenz (1.8 mg/kg i.p.) 24 h prior to heart isolation, and guanabenz (0.5 µM) was included in the perfusion buffer.

To induce AKI *in vivo*, animals were treated with two doses of tunicamycin (3 mg/kg/day i.p.) (Zinszner *et al*, 1998). Some animals were pre-treated with guanabenz (1.8 mg/kg i.p.). After sacrifice, kidneys were harvested for immunoblotting or analysis of apoptosis by TUNEL staining. Plasma urea concentration was measured using a commercial kit (Bioassay Systems).

### Statistics

Data are presented as mean ± SEM. Comparisons among groups were made by Student's *t*-test or one-way ANOVA and Tukey's *post-hoc* test, as appropriate. Kaplan–Meier analysis was used to compare survival. Statistical analyses were performed on GraphPad Prism (GraphPad Software, San Diego, CA). $P < 0.05$ was consid-ered significant.

Expanded View for this article is available online.

### Acknowledgements

This work was supported by the British Heart Foundation (RG/13/11/30384 [AMS], RE/13/2/30182 [AMS, CS]); a Fondation Leducq Transatlantic Network of Excellence Award (AMS); the Department of Health via a National Insti-tute for Health Research (NIHR) Biomedical Research Centre award to Guy's & St Thomas' NHS Foundation Trust in partnership with King's College London and King's College Hospital NHS Foundation Trust (AMS); a Norwe-gian Health Association Fellowship (ADH); a Russian Science Foundation grant 14-14-00747 (VB); and the German Research Foundation SFB 815 & 834 (KS & RPB). Microscopic images were acquired in the Nikon Imaging Centre at King's College London (Nic@King's), with support from John Harris. We thank John Pizzey for help with MEF isolation.

### Author contributions

CXCS designed and performed experiments, analyzed data, and wrote the paper. ADH, MB, MZ, CM, JK, DF, TVM, AMC, DM, MZS, NA, and KS performed experiments. CMS, ACB, RPB, EB, MP, VB, RC, RCH, and RAS designed and supervised experiments. AMS conceived the study, designed and supervised experiments, and wrote the paper. All authors edited the paper.

### Conflict of interest

The authors declare that they have no conflict of interest.

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
