## [Review Process File · The EMBO Journal]

Manuscript EMBO-2015-92394

Redox metal center inhibition of serine-threonine protein phosphatase 1 regulates eIF2 -mediated stress signaling

Celio Santos, Anne Hafstad, Matteo Beretta, Min Zhang, Chris Molenaar, Jola Kopec, Dina Fotinou, Thomas Murray, Andrew Cobb, Maira Zeh Silva, Daniel Martin, Narayana Anilkumar, Katrin Schröder, Catherine Shanahan, Alison Brewer, Ralf Brandes, Eric Blanc, Maddy Parsons, Vsevelod Belousov, Richard Cammack, Robert Hider, Roberto Steiner and Ajay Shah

Corresponding author: Ajay Shah, King's College London

Review timeline:

Submission date:	27 June 2015
Editorial Decision:	06 August 2015
Revision received:	08 November 2015
Accepted:	02 December 2015

Editor: Andrea Leibfried

Transaction Report:

1st Editorial Decision

06 August 2015

Thank you for submitting your manuscript entitled 'Redox metal center inhibition of serine-threonine protein phosphatase 1 regulates eIF2 α -mediated stress signaling'. I have now received reports from all referees, which are enclosed below.

As you will see, the referees appreciate your analyses and the diverse methodologies used to show a function of Nox4 in PP1 inhibition during the ER stress response. However, they also raise various concerns, which prevent publication in The EMBO Journal at this stage. The major concern is in regard to the specificity of PP1 inhibition towards eIF2 α , which is currently not conclusively shown as noted by both referee #1 and #3. This concern was also raised by an independent advisor with whom I discussed your manuscript. Furthermore, another major concern is that the kinetic data do not support your conclusions (referee #3, point 1).

Given the interest into the topic and the constructive comments provided by the referees, I can offer to consider a revised version should you be able to substantiate your model along the lines suggested by the referees. However, I would like to point out that I will need strong support from all referees on a revised version for further consideration here. Importantly, the above-mentioned issues regarding the specificity and the kinetics need to be addressed.

Furthermore,

- Referee #1's first point regarding the directionality needs to be discussed and I agree with this referee that the title should be changed.
- Referee #2's concerns should all be addressed (which should be straightforward)

- Referee #3's point 1 and point 2 are crucial to be addressed as indicated above. Point 3 of this referee can be addressed by discussion. The missing control mentioned in point 4 needs to be added and the discussion of the disease model data should be expanded.
- Referee #1 and #3 argue for a focus on Nox4, and I agree with this view.

I thus realize that carefully addressing all points raised by the referees demands a lot of work with uncertain outcome, so please consider your options carefully and let me know in case you do not want to embark into the revision. Please let me also know if you have any questions regarding the revision.

Thank you for the opportunity to consider your work for publication. I look forward to your revision.

REFEREE REPORTS

Referee #1:

The manuscript presented by Santos et al show a pro-survival role of the NADPH oxidase-4 (Nox4), involving the PERK-eIF2alpha-ATF4 branch of the UPR, in particular, ATF4 levels. They show that Nox4 deficiency decreases ER stress response protein levels, and also decreases eIF2alpha phosphorylation levels. They concluded through different approaches, including immunoprecipitation, gradient centrifugation and immunofluorescence assays, that Nox4 exerts its function interacting with GADD34-PP1 phosphatase complex, inhibiting the desphosphorylation activity. Also, they studied the possible mechanism involved in PP1 inhibition, and concluded that the metal binding site is involved in PP1 activity. They also performed an in vivo analysis, showing the protective role of Nox4 in cardiac injury, which is essential to increase the impact of the study

The manuscript presented by Santos et al is suitable for EMBO journal, with major points to addressed:

- The authors suggest a bidirectional positive signaling between Nox4 and ATF4.

But, there is no direct data or discussion about how ER stress impacts Nox4 function, or, does Nox4 function influences PERK-eIF2alpha-ATF4 pathway?

In particular, how does Nox4 activity influence ATF4?

The authors should included experimental data about Nox4 activity, in particular how does Nox4 activity directly impact or inhibit PP1.

- "Redox metal center inhibition of serine-threonine protein phosphatase 1 regulates eIF2a-mediated stress signaling". The title of the manuscript is over statement.

Although all the experiments are designed to demonstrate the participation of Nox4 in eIF2alpha phosphorylation, Nox4 does not appear on the title and it is only referred in one figure of the manuscript. The title should be changed and be centered around Nox4.

Referee #2:

The manuscript by Santos et al describes a mechanism for the regulation of eIF2alpha-mediated stress signalling through redox-dependent inhibition of PP1. The authors find that PP1 forms a complex with GADD34 and Nox4. Stress-induced activation of Nox4 results in H₂O₂ production and oxidation of the catalytic site-dinuclear metal centre Mn(II) and Fe(II). Oxidation of the metal centre is demonstrated by EPR. This mode of regulation differs from the cysteine-dependent oxidation of PTPs and DSPs. Crystal structures of oxidized and reduced PP1 shows contraction of metal-ligand distances in the oxidized state, consistent with oxidation of metal ions.

Overall this is a very interesting and important manuscript. The experiments were carefully

performed and controlled, and the authors employ an impressive range of techniques and approaches to understand cellular and molecular mechanisms. The manuscript certainly warrants publication in EMBO J.

Specific comments to consider.

1. 'catalytic activity' should be inserted before 'constitutively active' on page 4.
2. Page 8. The authors found that GADD34, PP1 and Nox4 co-eluted. Was this in the absence or presence of cellular stress?
3. P. 9. Define 'P437H-Nox4'.
4. The one experimental question relates to the metal-ligand bond distances for oxidized and reduced PP1. These decrease by between 0.1 to 0.2 Å. Since the resolution of the crystal structures is 2 Å, the variation in bond distance is very close to the coordinate error. The authors should comment on this. It would be useful to show difference electron density maps between the two structures to better visualize the change of metal-ligand structure.
5. Can the authors comment on why oxidation of Mn(II) and/or Fe(II) inactivates PP1?
6. Spelling should be consistent.

Referee #3:

The integrated stress response is characterized by the phosphorylation of eIF2 α by either the GCN or PERK kinase depending on the stress inducer. This inactivation of eIF2 α leads to a general inhibition of cap-dependent translation and simultaneously allows the ATF4-mediated induction of GADD34, the regulatory subunit of PP1. This complex mediates dephosphorylation of eIF2 α allowing translation to resume. This manuscript describes a role for Nox4, the ER localized member of the Nox family of NADPH oxidases, in regulating PP1 activity during ER stress. To summarize a very large amount of data, these investigators conclude that ER stress transcriptionally up-regulates Nox4 and that reducing or increasing Nox4 levels in turn modulates the UPR. Nox4 effects on the UPR are mediated through its ability to inhibit the phosphatase activity of PP1 specifically on eIF2 α , apparently due to the fact that Nox4, PP1 and GADD34 are in a complex. It would be argued that this would lead to more modest translational levels during the course of ER stress but this is not directly shown. Data are provided to argue that the induction of Nox4 during ER stress increases ER H₂O₂ levels, which in turn dramatically reduces the PP1 activity by oxidizing the bound metal ion. The relevance of this aspect of the ER stress response is addressed in an animal ischemia-reperfusion model and in response to acute in vivo kidney injury. They find that Nox4KO mice have more damage in both models, which can be partially normalized when combined with pharmacological antagonists of the GADD34:PP1 enzyme.

While the vast number of exceedingly diverse methodologies (ranging from enzymatic, biochemical, crystallographic, and animal assays) is impressive, it may also be part of the problem with this rather confusing manuscript. A number of the pieces of data are inconsistent with each other or with the literature. I've indicated some of these points below. In addition, the methods section and experimental details are too brief for the average reader who is unlikely to have expertise in all of these diverse methodologies. A more focused manuscript directed at more fully elucidating the mechanism of stress-induced, redox-regulated changes in PP1 activity seems critical before trying to understand complex animal models of disease.

Some specific points:

1. Experiments in Figure 1 and 2 show effects of the UPR on Nox4 and vice versa. The kinetics of the response are somewhat troubling, perhaps based on the time points shown and the stressor used. Increases in Nox4 protein are observed at 8 hrs, but eIF2 α phosphorylation is already decreasing at 4 hrs. If perhaps Nox4 is acting later in the response, why does shRNA to Nox4 prevent the initial phosphorylation of eIF2 α at the 2 hr time point well before it is induced? Perhaps using thapsigargin, a more acute ER stressor and more closely spaced early time points would shed some light on this issue. They state that there is no effect of Nox4 knock-down or over-expression on the ATF6 or Ire1 branches of the UPR. In spite of this, surprisingly Grp94, Grp78, and calnexin levels are down when Nox4 levels are suppressed and up when Nox4 is over-expressed. Under these

conditions translation should be enhanced or reduced respectively. Based on the published literature, in the latter case at least the levels of the chaperones should be reduced not increased. Why is this?

2. The effect of Nox4 on PP1 activity appears to be specific for eIF2 α , as other targets are not affected. What phosphopeptide is used in the experiments to measure phosphatase activity? Experiments argue that PP1, GADD34 and Nox4 are in a complex, presumably at the ER membrane and sucrose gradients suggest all three proteins sediment in similar fractions. Data are not presented to show that all PP1/GADD34 are at the membrane, which I doubt, or more importantly eIF2 α . Otherwise the argument of specificity for eIF2 α is hard to grasp mechanistically. Similarly why are the translation of the ATF4 transcription factor and in fact GADD34 affected unless their translation is localized as well. It simply wasn't clear what the authors were thinking/suggesting at this point.

3. They provide data to suggest that H₂O₂ is being produced in the ER by Nox4 in response to ER stress. Although I am not familiar with this assay the level of H₂O₂ generated looks very modest to my untrained eye. More information here would be useful. The cytosolic probes do not detect increases in H₂O₂, arguing I assume that it is contained within the ER lumen. They next set out to understand the mechanism of H₂O₂-induced suppression of PP1 activity, which is a cytosolic protein. PP1 is crystallized with and without added H₂O₂. They observe changes in the bond lengths between the metal ions and several interacting groups leading them to conclude that oxidation of the metal ions is responsible for changes in activity. While this evidence is rather circumstantial, it is indirectly supported by the fact that mutation of the single cysteine in PP1 does not ablate the H₂O₂ effects.

4. Finally two animal models are explored in an attempt to provide physiological relevance. While the data are intriguing the fact that many points of the mechanism of Nox4 regulation of PP1 activity are unclear makes it very hard to draw conclusions here. The inhibitors of PP1:GADD34 used here are only shown with the Nox4 KO mice, which do not even reduce injury to the level observed in the WT mice. There are no data for the inhibitor-treated WT mice, which should also fair better. Might this actually point to Nox4 effects that are independent of those on PP1 activity?

1st Revision - authors' response

08 November 2015

RESPONSE TO REFEREE #1

Thank you for the comments and suggestions which we have addressed as outlined in detail below. We hope that this satisfactorily addresses the points raised.

COMMENT #1: The authors suggest a bidirectional positive signaling between Nox4 and ATF4. But, there is no direct data or discussion about how ER stress impacts Nox4 function, or, does Nox4 function influences PERK-eIF2 α -ATF4 pathway? In particular, how does Nox4 activity influence ATF4?

RESPONSE:

We do indeed suggest bidirectional signaling between Nox4 and ATF4. First, the level of Nox4 (while already present basally) is significantly increased by ATF4 (Fig 1G&H and Fig S2C). We have now performed additional experiments to assess how ATF4 upregulates Nox4. Examination of the 10 kb rat genomic Nox4 promoter sequence proximal to the transcriptional start site identified 3 potential binding sites for ATF4. Using ChIP, we found that a region containing 2 canonical ATF4 binding motifs (-3525 to -3410, relative to the *Nox4* translational start site) demonstrated binding to ATF4, which increased in the presence of tunicamycin (Appendix Fig S2D). These results suggest that at least part of the increase in *Nox4* may involve direct cis-regulation by ATF4. The new data are reported on page 6 (para 2) and in Fig S2D.

Secondly, Nox4 itself upregulates ATF4. This is clear from the Nox4 knockdown and over-expression experiments in Fig 1C&D and the studies in MEFs in Fig 2I where we clearly demonstrate a dependence of the level of ATF4 upregulation on Nox4. The upregulation of ATF4 depends upon the phosphorylation of eIF2 α and our data suggest that Nox4 regulates ATF4 *indirectly* via changes in eIF2 α phosphorylation (Fig 2A,B). The mechanism by which Nox4

regulates eIF2 α phosphorylation is through the spatially-localized redox inhibition of PP1 activity, thereby enhancing eIF2 α phosphorylation - which is investigated in detail in the paper. We have now added further discussion to make the details of this bidirectional signaling clearer - please see Discussion bottom of page 16 to page 17.

COMMENT #2: The authors should included experimental data about Nox4 activity, in particular how does Nox4 activity directly impact or inhibit PP1.

RESPONSE:

The change (increase) in Nox activity in response to tunicamycin is shown in Fig S1D. We also show with the use of multicolor imaging that there is a localized Nox4-dependent increase in ROS at the ER after tunicamycin (Fig 3G). The mechanism by which Nox4 inhibits PP1 involves a molecular association between GADD34 and Nox4 (Fig 3D-F, Fig EV1A-C), which facilitates high localized ROS generation adjacent to PP1 (Fig 3G). We also show in experiments with the Nox4(P437H) mutant that Nox4 enzymatic activity is required to inhibit eIF2 α -targeted PP1 (Fig 3H and Fig EV1F). Finally, we show the redox sensitivity of the Nox4 effect on PP1 in the experiments reported in Fig 4F, where the impact of ascorbate on the Nox4-dependent response is studied. At the level of PP1, our studies suggest oxidation of the metal center of the enzyme as the mechanism for inhibition.

Taken together, we hope that these set of studies provides a clear and detailed picture of how Nox4 influences eIF2 α -targeted PP1 activity.

COMMENT #3: The title of the manuscript is over statement. Although all the experiments are designed to demonstrate the participation of Nox4 in eIF2alpha phosphorylation, Nox4 does not appear on the title and it is only referred in one figure of the manuscript. The title should be changed and be centered around Nox4.

RESPONSE:

We have changed the title of the paper to focus more on Nox4. Most of the figure legends now refer to Nox4.

RESPONSE TO REFEREE #2

Thank you for the comments and suggestions which we have addressed as outlined in detail below. We hope that this satisfactorily addresses the points raised.

COMMENT #1. 'catalytic activity' should be inserted before 'constitutively active' on page 4.

RESPONSE: This has been done.

COMMENT #2. Page 8. The authors found that GADD34, PP1 and Nox4 co-eluted. Was this in the absence or presence of cellular stress?

RESPONSE:

The results in Fig 3C were in tunicamycin-stimulated cells. This has been stated in the Figure legend. We have now also included additional experiments in membrane and cytosolic fractions (Fig 3B) which show the time course of stress-induced membrane-enrichment of GADD34, PP1, Nox4 and eIF2 α .

COMMENT #3. P. 9. Define 'P437H-Nox4.

RESPONSE:

This has now been done: "a mutant Nox4 construct with a single proline to histidine amino acid substitution at residue 437 in the NADPH-binding domain, P437H-Nox4" - page 10, para 3.

COMMENT #4. The one experimental question relates to the metal-ligand bond distances for oxidized and reduced PP1. These decrease by between 0.1 to 0.2 Å. Since the resolution of the

crystal structures is 2 Å, the variation in bond distance is very close to the coordinate error. The authors should comment on this. It would be useful to show difference electron density maps between the two structures to better visualize the change of metal-ligand structure.

RESPONSE:

The relevant analysis is shown in the Appendix pages 24-25 and Fig EV2E (this was suppl Fig 5f in the original submission). Essentially, it shows that there is a statistically significant correlation for the average metal-coordination contraction upon oxidation between the crystallographic experiment and theoretical calculations (Zhang *et al.* 2013, cited in the paper). Prompted by the reviewer's comment, we have realized that Fig 4d in the main body of the original submission put excessive and partly misleading emphasis on the *individual* metal-ligand distances. The strength of the analysis derives from the analysis of *average* metal-coordination distances and the comparison with theoretical prediction (Appendix pages 24-25). We have therefore removed this panel in the revised manuscript and replaced it with supplementary Fig 5e of the original submission. No changes have been introduced in the text and all metal-ligand distances are available in Table S1.

The 'difference electron density map between the structures' suggested by the reviewer unfortunately cannot be calculated as the data sets are not isomorphous (please see unit cell parameters in Table S3).

COMMENT #5. Can the authors comment on why oxidation of Mn(II) and/or Fe(II) inactivates PP1?

RESPONSE:

The theoretical study of Zhang *et al.* 2013 predicts that oxidation of Mn centers in PP1 leads to shrinkage of the coordination sphere with a resulting increase in the energy barrier for phosphate hydrolysis. This is consistent with an inhibition effect, and different from the PTP1 inhibition mechanism where phosphate binding is inhibited. The oxidation of Fe(II) to Fe(III) is expected to have a similar shrinkage effect, since the ionic radius of the metal would shrink by a similar amount upon the loss of an electron (Cotton *et al.*, 1999, cited in the paper).

We have discussed this in the Results page 12 and the Discussion page 18, para 2.

COMMENT #6. Spelling should be consistent.

RESPONSE: Apologies - we have now tried to be consistent in the use of American spelling.

RESPONSE TO REFEREE #3

Thank you for the comments and suggestions which we have addressed as outlined in detail below. We hope that this satisfactorily addresses the points raised.

GENERAL COMMENTS: The methods section and experimental details are too brief for the average reader who is unlikely to have expertise in all of these diverse methodologies.

RESPONSE:

We have substantially expanded the Materials and Methods section and also provided detailed methods in the Appendix.

SPECIFIC POINT #1: Experiments in Figure 1 and 2 show effects of the UPR on Nox4 and vice versa. The kinetics of the response are somewhat troubling, perhaps based on the time points shown and the stressor used. Increases in Nox4 protein are observed at 8 hrs, but eIF2α phosphorylation is already decreasing at 4 hrs. If perhaps Nox4 is acting later in the response, why does shRNA to Nox4 prevent the initial phosphorylation of eIF2α at the 2 hr time point well before it is induced? Perhaps using thapsigargin, a more acute ER stressor and more closely spaced early time points would shed some light on this issue.

They state that there is no effect of Nox4 knock-down or over-expression on the ATF6 or Ire1 branches of the UPR. In spite of this, surprisingly Grp94, Grp78, and calnexin levels are down when Nox4 levels are suppressed and up when Nox4 is over-expressed. Under these conditions translation should be enhanced or reduced respectively. Based on the published literature, in the latter case at least the levels of the chaperones should be reduced not increased. Why is this?

RESPONSE:

Kinetics: It is important to note that Nox4 is already present basally, albeit at relatively low level as compared to after the induction of ER stress (e.g. Fig 1b, 1d, 1h, 2a, 2b of the original version). In addition, Nox4 has susceptibility to ubiquitination and proteasomal degradation (see for instance recent paper by Desai et al, J Biol Chem 2014;289:18270-8, now cited in the manuscript) so that its detection at lower levels requires care to avoid degradation during sample preparation. We followed the reviewer's suggestion to study the kinetics of increase in Nox4 levels after exposure to tunicamycin more carefully and we also looked at the response to thapsigargin, as recommended. We added a proteasome inhibitor (Mg132) immediately before cell harvesting in order to minimize Nox4 degradation. These results are shown in the new Fig 1A and 1B and indicate that Nox4 levels rise very rapidly after the addition of either stressor. In new ChiP experiments performed in response to the suggestion of reviewer #1, we find that at least part of the increase in Nox4 may be transcriptionally mediated via ATF4 (Appendix Fig S2D). However, there might be additional mechanisms - such as changes in protein stability - that contribute to the very early increase in levels. We have now discussed this point further on page 17, para 2.

Levels of ER chaperones: Previous work has shown that ER chaperones such as Grp78 have ATF4 binding sites that regulate their expression during stress (e.g. Luo et al., JBC, 2003, now cited in the paper) and it has also been shown that ATF4 KO cells or GADD34 KO cells show reduced stress-induced expression of ER chaperones (Ma and Hendershot, 2003; Kojima et al, 2003; both cited in the paper). Therefore, the response to Nox4 perturbation (increased chaperones after Nox4 overexpression and decreased chaperones after Nox4 knockdown) is consistent with these prior data. We further confirmed this by looking at the levels of Grp78 and Grp94 in cells in which ATF4 was knocked down, using an identical approach to that shown in Fig 1G. As shown in the figure below, the stress-induced increases in Grp78 and Grp94 are much lower after ATF4 knockdown as compared to control.

We have now mentioned in the Discussion the fact that ER stress is accompanied by an increase in ER chaperones and referenced the papers by Luo et al (JBC, 2003) and Ma and Hendershot, 2003 - please see first paragraph of the Discussion on page 16.

Refs

- Luo S et al (2003) Induction of Grp78/BiP by translational block. J Biol Chem 278: 37375-37385.
- Kojima E et al (2003) The function of GADD34 is a recovery from a shutoff of protein synthesis induced by ER stress: elucidation by GADD34-deficient mice. FASEB J 17: 1573-1575
- Ma Y, Hendershot LM (2003) Delineation of a negative feedback regulatory loop that controls protein translation during endoplasmic reticulum stress. J Biol Chem 278: 34864-34873

SPECIFIC POINT #2. The effect of Nox4 on PPI activity appears to be specific for eIF2 α , as other targets are not affected. What phosphopeptide is used in the experiments to measure phosphatase activity? Experiments argue that PPI, GADD34 and Nox4 are in a complex, presumably at the ER membrane and sucrose gradients suggest all three proteins sediment in similar fractions. Data are not presented to show that all PPI/GADD34 are at the membrane, which I doubt, or more importantly eIF2 α . Otherwise the argument of specificity for eIF2 α is hard to grasp mechanistically. Similarly why are the translation of the ATF4 transcription factor and in fact GADD34 affected unless their translation is localized as well. It simply wasn't clear what the authors were thinking/suggesting at this point.

RESPONSE: We used the phospho-threonine peptide (Lys-Arg-phosphoThr-Iso-Arg-Arg, K-R-pT-I-R-R). This is now stated in the Materials and Methods page 26, para 3.

The reviewer makes an important point about localization of PP1, GADD34 and eIF2 α at the membrane. The targeting of PP1 to eIF2 α has long been known to be determined by GADD34 but very recent work indicates that GADD34 interacts with *both* PP1 and eIF2 α at the ER, thereby acting as a scaffold (Choy et al 2015, now cited in the paper). This new study provides a further rationale for the specificity of Nox4/GADD34/PP1 action towards eIF2 α , mediated by the interaction of Nox4 with GADD34. We have now undertaken new experiments to look at the relative distributions of Nox4, GADD34, PP1 and eIF2 α in membrane and cytosolic fractions prepared from tunicamycin-treated H9c2 cells. Following treatment with tunicamycin, there was a marked enrichment of all 4 proteins in the membrane fraction (Fig 3B), consistent with their recruitment and co-localization. The membrane fraction is enriched in ER proteins (Appendix Fig S4A). We have also investigated the presence of eIF2 α in the sucrose gradient centrifugation experiments. This showed that a significant proportion of eIF2 α co-eluted with GADD34, PP1 and Nox4 (Fig 3C), providing further evidence for a significant co-localization of these proteins. (Not all eIF2 α co-eluted, as predicted by the reviewer.) These results together with the prior data that was already included in the paper provide a strong rationale and mechanistic basis for the specific modulation by Nox4 of eIF2 α phosphorylation, as opposed to other PP1 targets. The new data are presented in Fig 3B&C and on page 9, para 2 of the Results.

With respect to protein translation, this is increasingly recognized to be a quite complex process during ER stress. It is very well established that while the phosphorylation of eIF2 α during the UPR reduces the translation of many proteins (especially those that are folded in the ER), the translation of other proteins such as ATF4 increases - related to a cap-independent translation (e.g. Vattem and Wek, 2004, cited in the paper). In other words, an increase in translation of ATF4 is readily understood and entirely consistent with prior literature. The *site* of ATF4 translation within the cell is not clear as far as we are aware. However, a recent comprehensive translational profiling analysis of cells subjected to protein unfolding stress published in Cell (Reid et al, Cell 2014) reports that (a) there are directionally opposite changes in the translation of different subsets of proteins during the UPR (please see Fig 2 in that paper) and (b) the subcellular localization of mRNAs and translation can change quite dynamically. This is an interesting area for further study but well beyond the scope of the current manuscript. We have emphasised in the first paragraph of the Discussion on page 16 that while eIF2 α phosphorylation results in a reduction in synthesis of most ER proteins, there is an increase in translation of some proteins such as ATF4.

Refs

- Choy MS et al (2015) Structural and functional analysis of the GADD34:PP1 eIF2 α phosphatase. Cell Reports 11: 1885–1891
- Reid DW, Chen Q, Tay AS, Shenolikar S, Nicchitta CV (2014) The unfolded protein response triggers selective mRNA release from the endoplasmic reticulum. Cell 158: 1362-1374

SPECIFIC POINT #3. They provide data to suggest that H2O2 is being produced in the ER by Nox4 in response to ER stress. Although I am not familiar with this assay the level of H2O2 generated looks very modest to my untrained eye. More information here would be useful.

RESPONSE: The quantitation of Hyper fluorescence is shown in Fig EV1 D&E. The response to extracellular H₂O₂ (200 nM) was used as a positive control. This is now better explained in the Methods and the legend to Fig 3. The level of increase in fluorescence observed after tunicamycin treatment is comparable to increases reported in previous studies that investigated localized redox signaling using such probes (e.g. Ermakova YG et al Nature Commun 2014, cited in the paper; Belousov VV et al Nature Methods 2006, cited in the Appendix).

SPECIFIC POINT #4. Finally two animal models are explored in an attempt to provide physiological relevance. While the data are intriguing the fact that many points of the mechanism of Nox4 regulation of PPI activity are unclear makes it very hard to draw conclusions here. The inhibitors of PPI:GADD34 used here are only shown with the Nox4 KO mice, which do not even reduce injury to the level observed in the WT mice. There are no data for the inhibitor-treated WT mice, which should also fair better. Might this actually point to Nox4 effects that are independent of those on PPI activity?

RESPONSE:

We have now added the experiments showing the effect of guanabenz on WT mouse hearts subjected to ischemia/reperfusion (Fig 5D). Guanabenz had no effect on infarct size in WT hearts suggesting that the Nox4-dependent regulation of eIF2 α /ATF4 signaling cannot be further enhanced in normal hearts during acute I/R and that other mechanisms may also contribute to cardiac injury. However, in the setting of acute kidney injury, guanabenz improved outcome both in Nox4 KO and WT mice, indicating that protective eIF2 α /ATF4 signaling can be further boosted during AKI even in normal mice with competent Nox4 regulation.

As recommended by the Editor, we have now expanded the Discussion of these results in disease models (page 19 last para through to page 20). We have also emphasised that the main purpose of these experiments is to provide provide proof-of-concept for the existence and importance of Nox4-dependent regulation of eIF2 α /ATF4 signaling during acute heart and kidney injury, whereas potential therapeutic manipulation would require detailed assessment in future studies.

2nd Editorial Decision

02 December 2015

Thank you for sending your revised manuscript to us. I have received now all reports on your manuscript, and I am pleased to inform you that your manuscript has been accepted for publication in the EMBO Journal.

REFEREE REPORTS

Referee #1:

In this version of the manuscript, the authors have addressed all the suggestions and questions made. They developed the experiments (ChIP assays) to show that ATF4 can bind Nox4 promoter and also included more detail discussion about Nox4 and ATF4 bi directional signaling. The title was modified, including "Nox4" in it. Now the paper is suitable to be published in EMBO journal.

Referee #2:

The authors have addressed my comments and the manuscript now warrants publication in EMBO Journal.

Referee #3:

The authors have responded suitably to the points I raised and have provided additional data to support their claims. I still feel this manuscript has some weaknesses in terms of the conclusions drawn (for instance the kinetics of Nox4 induction versus the responses it controls), but I do not think these are sufficiently troublesome to block its publication.